# Inactivation of NF-κB2 (p52) restrains hepatic glucagon response via preserving PDE4B induction

Wen-Song Zhang[1], An Pan[1], Xu Zhang[1], Ang Ying[1], Gaoxiang Ma[1,2], Bao-Lin Liu[1], Lian-Wen Qi [1,2]*, Qun Liu [1]* & Ping Li[1]*

Glucagon promotes hepatic gluconeogenesis and maintains whole-body glucose levels during fasting. The regulatory factors that are involved in fasting glucagon response are not well understood. Here we report a role of p52, a key activator of the noncanonical nuclear factor-kappaB signaling, in hepatic glucagon response. We show that p52 is activated in livers of HFD-fed and glucagon-challenged mice. Knockdown of p52 lowers glucagon-stimulated hyperglycemia, while p52 overexpression augments glucagon response. Mechanistically, p52 binds to phosphodiesterase 4B promoter to inhibit its transcription and promotes cAMP accumulation, thus augmenting the glucagon response through cAMP/PKA signaling. The anti-diabetic drug metformin and ginsenoside Rb1 lower blood glucose at least in part by inhibiting p52 activation. Our findings reveal that p52 mediates glucagon-triggered hepatic gluconeogenesis and suggests that pharmacological intervention to prevent p52 processing is a potential therapeutic strategy for diabetes.

[1] State Key Laboratory of Natural Medicines, School of Traditional Chinese Pharmacy, China Pharmaceutical University, Nanjing 210009, China. [2] Clinical Metabolomics Center, China Pharmaceutical University, Nanjing 211198, China. *email: Qilw@cpu.edu.cn or liuquncpu@126.com or liping2004@126.com

Insulin and glucagon are responsible for maintaining whole-body glucose homeostasis. Insulin reduces postprandial hyperglycemia by promoting glucose disposal to target tissues, while glucagon stimulates hepatic gluconeogenesis to maintain blood glucose levels during fasting or starvation. Accumulating evidence from human and animal studies shows that plasma glucagon concentrations are abnormally elevated in individuals with obesity and/or diabetes[1,2]. Dysregulation of the hepatic glucagon response induces excessive hepatic glucose production, contributing to fasting hyperglycemia in diabetes[3,4]. Mechanisms to reduce circulating glucagon levels and antagonize the hepatic glucagon response in target tissues are well-recognized means to reducing hyperglycemia in diabetes[5–8].

The hepatic glucagon response is mediated by activation of the cyclic AMP (cAMP)/protein kinase A (PKA) pathway. By binding to the G-protein-coupled receptor, glucagon activates adenylyl cyclase signaling. This leads to production of cAMP, which activates PKA and induces phosphorylation of cAMP-response element-binding protein (CREB). After phosphorylation, CREB promotes gluconeogenesis through induction of glucose-6-phosphatase (G6Pase), phosphoenolpyruvate carboxykinase (PEPCK), and peroxisome proliferator-activated receptor gamma coactivator-1 alpha (PGC-1α). Because cAMP acts as a second messenger to initiate downstream signaling cascades in the gluconeogenic pathway, regulation of cAMP is a key step in the control of hepatic glucagon response. Inhibition of adenylyl cyclase enzymes reduces hepatic glucose production by suppressing hepatic glucagon signaling[8]. cAMP can be degraded by phosphodiesterases (PDEs) to prevent excessive accumulation. Metabolic alterations could affect PDEs activity and then influence hepatic gluconeogenesis. AMP-activated protein kinase (AMPK) and hypoxia-inducible factor 2α increase PDEs activity and thus antagonize hepatic glucagon-stimulated cAMP signaling[9,10]. However, the intracellular mechanisms that regulate glucagon-mediated responses are incompletely understood.

The nuclear factor-kappa B (NF-κB) family of transcription factors includes NF-κB1 (p50), NF-κB2 (p52), RelA (p65), RelB, and c-Rel, and these regulate diverse cellular processes[11]. Various extracellular signals activate NF-κB intracellular signaling, which results in translocation of DNA-binding heterodimers and homodimers to the nucleus and transcriptional activation of target genes. Canonical NF-κB1[12,13] signaling is well recognized as the central regulator of inflammatory responses, but the role of NF-κB2 in human diseases is relatively less studied. Roles of p52 in lymphoid organogenesis, B-cell maturation, and osteoclast differentiation have only recently been appreciated[14,15], and aberrant overexpression of p52 has been reported in genetic and nutritionally obese mice[16]. However, the role of p52 in hepatic gluconeogenesis remains largely unknown.

To know if NF-κB2 activation is involved in metabolic disorders, this study aims to investigate the role of p52 activation in hepatic gluconeogenesis. Our work shows that p52 binds to PDE4B promoter to inhibit its transcription and promotes cAMP accumulation, thus augmenting the glucagon response through cAMP/PKA signaling. We also show that metformin and ginsenoside Rb1 restrain hepatic glucagon response through a p52-dependent manner.

## Results

**Hepatic NF-κB2 is abnormally activated in obese humans.** In search of a correlation between hepatic *NF-κB2* expression and BMI, the RNAseq data of the Genotype-Tissue Expression Project (GTEx) were downloaded from the Genotype and Phenotypes (dbGaP, phs000424.v7.p2) database[17]. As shown in Supplementary Fig. 1a, *NF-κB2* expression in liver samples of 51 obese

individuals correlated positively with BMI ($r = 0.36$, $p = 0.01$). In close agreement, when we examined livers of fasted high-fat diet (HFD)-fed mice, we observed a significant increase in p52 activation compared with those of chow-fed mice (Supplementary Fig. 1b).

**p52 mediates HFD-induced hepatic gluconeogenesis.** To investigate the role of p52 in hyperglycemia in vivo, we fed mice a HFD and silenced p52 using a siRNA transfection technology (Fig. 1a). HFD feeding resulted in fasting hyperglycemia (Fig. 1b) and impaired oral glucose tolerance (Supplementary Fig. 2a). We found that p52 knockdown lowered fasting blood glucose levels (Fig. 1b) and improved oral glucose tolerance (Supplementary Fig. 2a) in HFD-fed mice. Pyruvate provides a substrate for hepatic gluconeogenesis, and is an indicator of hepatic glucose production. HFD-fed mice had increased blood glucose levels in response to a pyruvate load compared with normal chow diet (NCD)-fed mice, but p52 knockdown reversed the glucose increase (Fig. 1c). As expected, fasting serum glucagon levels were increased in HFD-fed mice. p52 knockdown did not impact glucagon secretion in HFD-fed mice (Fig. 1d), suggesting that p52 may regulate glucagon signaling rather than glucagon secretion. Of note, p52 knockdown resulted in reduced body weight gain, less fat mass, and decreased lipid deposition in the livers of mice fed with HFD for 8 weeks (Supplementary Fig. 2b–d). Food intake was not changed by p52 knockdown (Supplementary Fig. 2b).

To observe the impact of p52 on glucagon response, we employed an acute glucagon-challenged mice model. After overnight fasting, hepatic glycogen was breakdown completely, and glucagon was not able to activate glycogenolysis (data not shown). We observed that p52 knockdown attenuated glucagon-stimulated hyperglycemia (Fig. 1e). In addition, we used AAV8-shRNA to establish liver-specific p52 knockdown mice (Supplementary Fig. 3a) for further confirmation. The results in liver-specific knockdown mice mirrored the results in siRNA transfection mice (Supplementary Fig. 3b, c). Along the same line, we used hepatotropic AAV-8 as vehicle to specifically overexpress p52 in the liver (Supplementary Fig. 3d). The results showed that p52 liver-specific overexpression increased fasting blood glucose (Fig. 1f) and augmented glucagon-stimulated hyperglycemia in mice (Fig. 1g).

**p52 knockdown blocks cAMP/PKA signaling.** Glucagon-induced hepatic gluconeogenesis relies on cAMP/PKA signaling. HFD feeding increased hepatic cAMP accumulation, whereas knockdown of p52 reduced cAMP accumulation and effectively prevented PKA activation (Fig. 2a, b). In response to PKA activation, CREB was phosphorylated to upregulate key gluconeogenesis-associated enzymes. HFD increased CREB phosphorylation (Fig. 2c) and upregulated transcription of *G6Pase*, *PEPCK*, and *PGC-1α* (Fig. 2d). Knockdown of p52 inactivated CREB through dephosphorylation (Fig. 2c), and reversed gluconeogenesis-associated genes alterations (Fig. 2d). In close agreement, glucagon challenge increased p52 expression (Fig. 2e), stimulated cAMP accumulation (Fig. 2f), activated PKA (Fig. 2g), phosphorylated CREB (Fig. 2h), and increased *G6Pase*, *PEPCK*, and *PGC-1α* mRNA expression in mice (Fig. 2i; Supplementary Fig. 3c). Correspondingly, p52 siRNA transfection reversed these alternations in the liver of glucagon-challenged mice. These results showed that inactivation of p52 restrained hepatic glucagon response.

**p52 inhibits PDE4B expression to promote cAMP accumulation.** To investigate the underlying mechanism by which p52

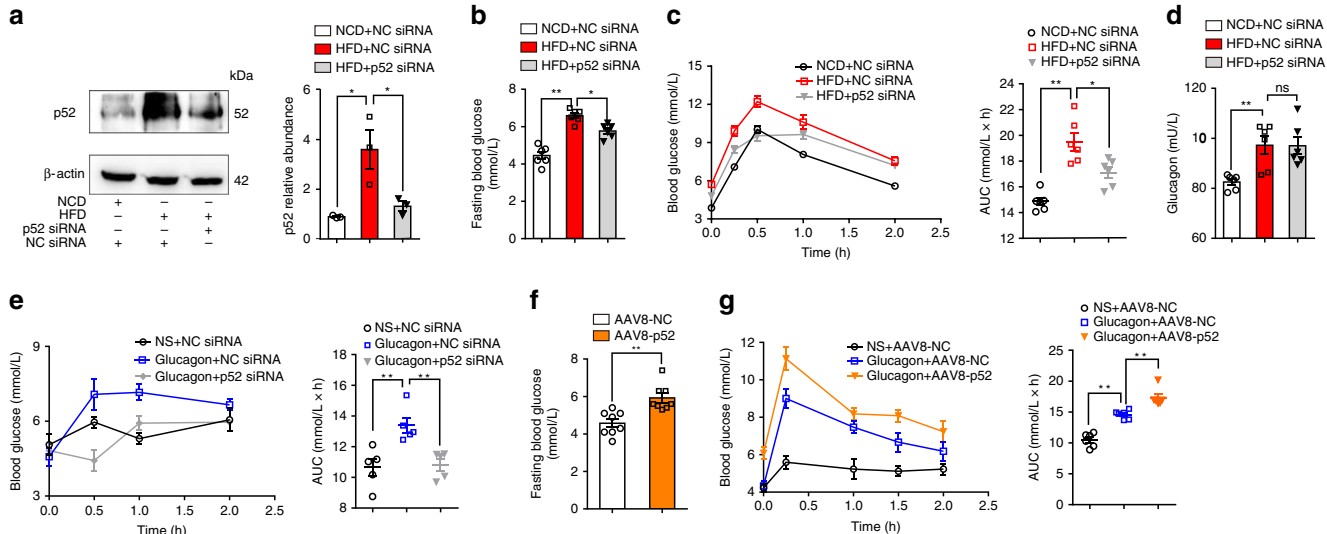

**Fig. 1** Treatment with p52 siRNA reduces fasting hyperglycemia in HFD-fed mice. **a** p52 protein level in liver tissue of NCD-fed, HFD-fed, and HFD-fed mice treated with p52 siRNA or NC siRNA. Liver tissues were collected from the mice after 8 weeks feeding ($n = 3$). **b** Fasting blood glucose in mice in panel **a** after 7 weeks feeding ($n = 6$). **c** Pyruvate tolerance test (2 g/kg body weight) in the mice in panel **a** after 7 weeks of feeding. AUC is indicated on the right ($n = 6$). **d** Fasting serum glucagon levels in mice in panel **a** fed with NCD or HFD for 8 weeks ($n = 6$). **e** Blood glucose levels in normal mice subjected to glucagon challenge (2 mg/kg body weight) and treatment with p52 siRNA or NC siRNA. AUC is indicated on the right ($n = 6$). **f** Fasting blood glucose of liver-specific p52 overexpression mice ($n = 8$). **g** Blood glucose curve and AUC for mice that are either treated with AAV8-p52 or AAV8-NC after glucagon injection ($n = 6$). AAV adeno-associated virus, NCD normal chow diet, HFD high-fat diet, AUC area under the curve, NS normal saline. Bars represent mean ± SEM values. Statistical difference in panel **f** was determined by a two-tailed Student's *t* test, and all others were used one-way ANOVA. *$p < 0.05$ vs. the control group, **$p < 0.01$ vs. the control group. Source data are provided as a Source Data file

mediates glucagon response, we focused on phosphodiesterase family. PDE4B is the predominant isoform of phosphodiesterase responsible for cAMP degradation in the liver. HFD and glucagon challenge inhibited PDE4B expression in the livers of mice, and this suppression was prevented by p52 knockdown (Fig. 3a, b). We then investigated the effect of p52 activation on the regulation of PDE4B and cAMP in HepG2 cells. In response to glucagon stimulation, PDE4B expression at the mRNA (Fig. 3c) and protein levels (Fig. 3d) decreased with cAMP accumulation (Fig. 3e), whereas silencing of p52 with siRNA preserved PDE4B expression and effectively ameliorated glucagon-stimulated cAMP production (Fig. 3c–e). In contrast, overexpression of p52 enhanced glucagon-stimulated reduction in PDE4B mRNA (Fig. 3f) and protein levels (Fig. 3g), and promoted cAMP production (Fig. 3h).

To verify whether or not p52 silencing antagonizes glucagon signaling was dependent on PDE4B, p52 siRNA and PDE4B siRNA were co-transfected in primary hepatocytes (Supplementary Fig. 4a, b). We observed that the inhibitory effects of p52 silencing were blocked by p52 siRNA and PDE4B siRNA co-transfection (Fig. 3i), indicating that p52 silencing inhibited gluconeogenesis in a PDE4B-dependent manner.

PDE3B is also expressed in the liver and is responsible for cAMP degradation. However, when stimulated by glucagon, PDE3B mRNA expression levels did not change significantly in vitro or in vivo (Fig. 3j). These results showed that p52 activation selectively suppressed PDE4B induction to increase cAMP accumulation in response to glucagon stimulation.

**p52 binds to PDE4B promoter and reduces its transcription.** As a transcription factor, p52 regulates gene expression through interaction with the promoter DNA; therefore, we hypothesized that p52 regulated PDE4B gene expression by interacting with its promoter. Western blotting and immunofluorescence confocal microscopy assays showed that glucagon promoted p52 nuclear

translocation (Fig. 4a, b). We tested the function of p52 on PDE4B promoter expression by a luciferase reporter assay. The results showed that the PGL3-basic-PDE4B promoter activity was inhibited by p52 co-transfection (Fig. 4c), indicating that p52 inhibited PDE4B transcription by interacting with its promoter. To further explore the impact of p52 on PDE4B transfection, we performed ChIP assays. We found two potential κB-binding sites in the PDE4B promoter region. The association of p52 at PDE4B promoter site A was 10.8-fold higher and at site B was 8.8-fold higher in glucagon-stimulated cells than in control-treated cells (Fig. 4d, e).

**Glucagon induced p52 activation by the cAMP/PKA pathway.** To explore the underlying mechanisms by which glucagon induced p52 activation, we stimulated primary hepatocytes with glucagon. We observed that the increase in p52 protein levels was prevented by the glucagon receptor inhibitor adomeglivant (Supplementary Fig. 5a). Forskolin activates adenylyl cyclase to generate cAMP from ATP. Bt2-cAMP is typically used to mimic cellular cAMP. Similar to stimulation by glucagon, forskolin and Bt2-cAMP increased p52 protein levels in hepatocytes (Supplementary Fig. 5b, c). H89, a PKA inhibitor, inhibited p52 expression stimulated by glucagon, forskolin, and Bt2-cAMP (Supplementary Fig. 5b–d). H89 also abrogated glucagon-induced p100 phosphorylation increase and p100 protein levels decrease (Supplementary Fig. 5e). Moreover, when hepatocytes stimulated with glucagon for different times, p100 decreased in a time-dependent manner while p52 rise accordingly (Supplementary Fig. 5f). But when pretreated with MG132, a cell-permeable proteasome inhibitor, p100 remained unchanged (Supplementary Fig. 5g). These results indicated that glucagon induced proteasome-mediated cleavage of p100 to p52. Interestingly, the transcriptional levels of NF-κB2 by glucagon stimulation was increased (Supplementary Fig. 5h), but diminished when pretreated with MG132 (Supplementary Fig. 5i).

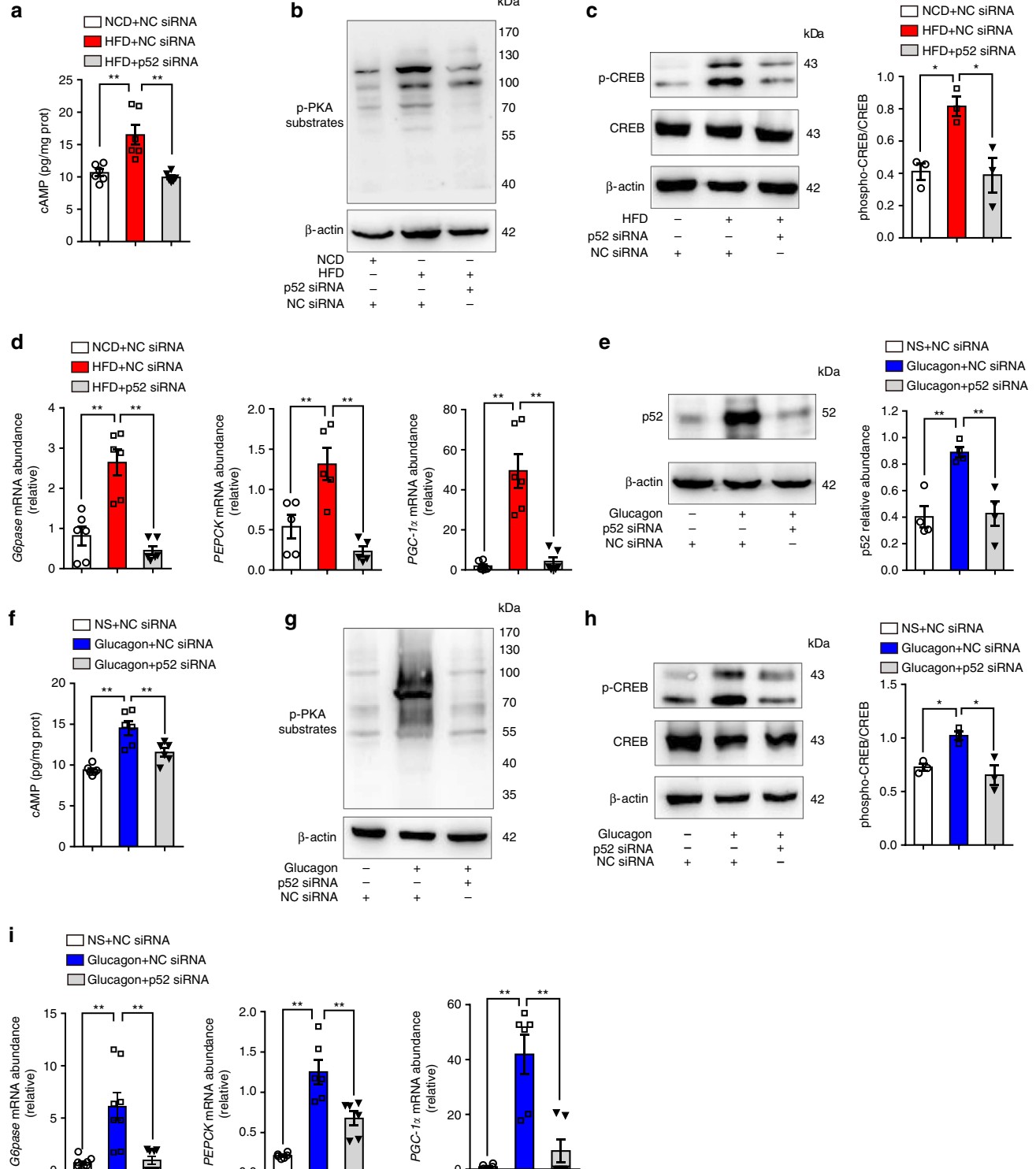

**Fig. 2** p52 knockdown blocks cAMP/PKA signaling. **a** Hepatic cAMP accumulation in the liver tissue of NCD-fed, HFD-fed, and HFD-fed mice with p52 silencing. Liver tissues were collected from the mice after 8 weeks feeding ($n = 6$). **b**, **c** Western blot analysis of phospho-PKA substrates (**b**) and p-CREB (**c**) expression in the liver tissue from the mice in panel **a** ($n = 3$). **d** qRT-PCR determination of mRNA levels of *G6pase*, *PEPCK*, and *PGC-1α* in the livers from the mice in panel **a** ($n = 6$). **e** Western blot analysis of p52 expression using lysates of the liver tissue from normal mice treated with glucagon (2 mg/kg body weight) and p52 siRNA or NC siRNA ($n = 4$). **f** Hepatic cAMP accumulation in the liver tissue of the mice ($n = 6$). **g**, **h** Hepatic phospho-PKA substrates and p-CREB protein levels in the mice ($n = 3$). **i** Hepatic mRNA levels of *G6pase*, *PEPCK*, and *PGC-1α* in the mice ($n = 6$). NCD normal chow diet, HFD high-fat diet, PKA protein kinase A, CREB cAMP-response element-binding protein, qRT-PCR quantitative real-time polymerase chain reaction, NS normal saline, *G6pase* glucose-6-phosphatase, *PEPCK* phosphoenolpyruvate carboxykinase, *PGC-1α* peroxisome proliferator-activated receptor gamma coactivator-1 alpha. Each bar represents mean ± SEM values. Statistical differences were determined by one-way ANOVA. *$p < 0.05$ vs. the control group, **$p < 0.01$ vs. the control group. Source data are provided as a Source Data file

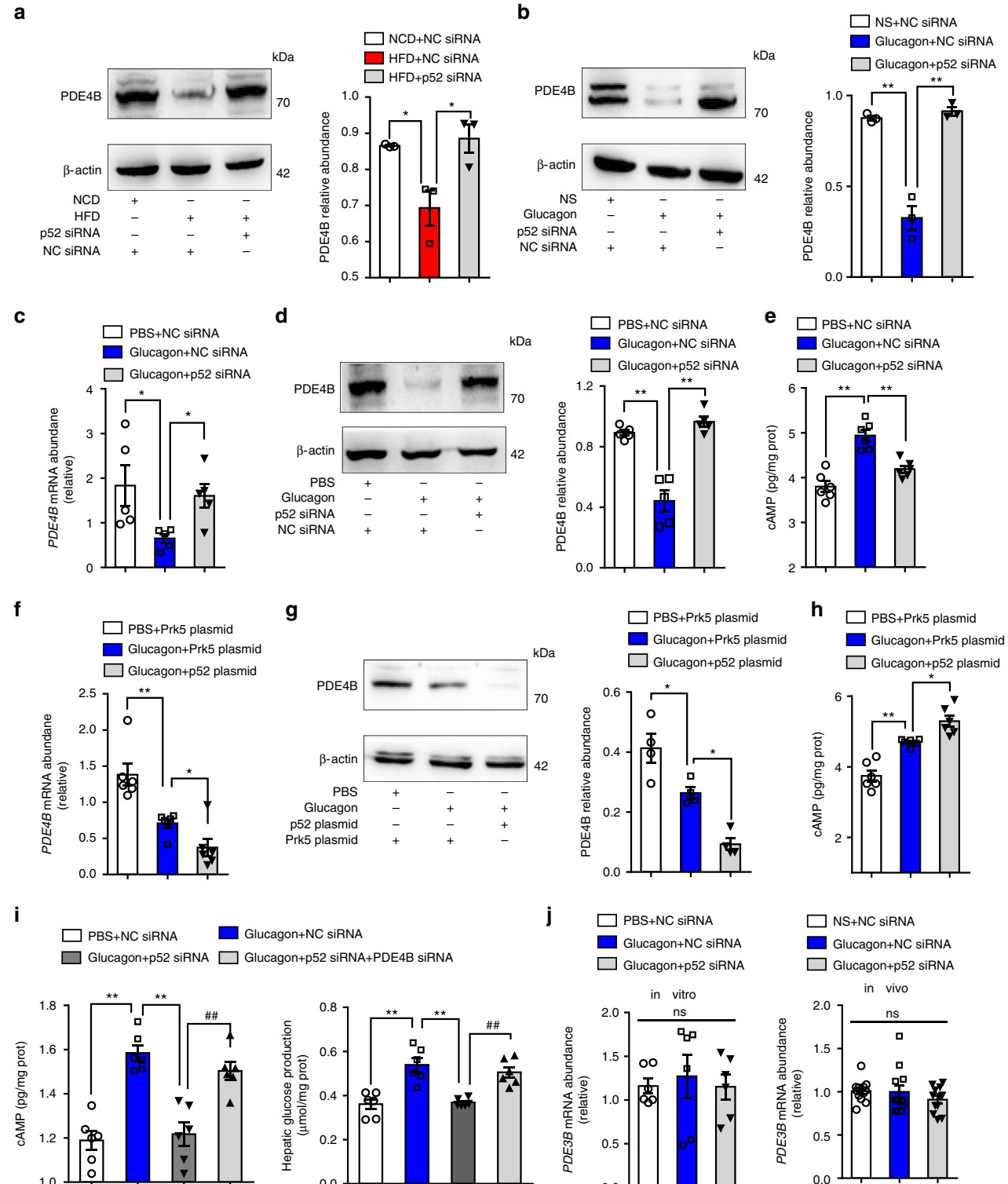

**Fig. 3** p52 inhibits PDE4B expression to promote cAMP accumulation. **a**, **b** Liver PDE4B protein expression in HFD-fed (**a**) and glucagon-stimulated mice (**b**). Bar graphs represent data normalized to β-actin levels ($n = 3$). **c** The mRNA levels of *PDE4B* in HepG2 cells transfected with p52 or NC siRNA ($n = 5$). Bar graphs represent the levels of genes normalized to *β-actin*. **d** The protein expression of PDE4B in p52 knocked down HepG2 cells, β-actin levels served as loading control ($n = 5$). **e** cAMP level in HepG2 cells transfected with p52 siRNA ($n = 6$). **f** The mRNA levels of *PDE4B* in HepG2 cells transfected with p52 overexpression plasmid ($n = 5$). **g** Western blotting of PDE4B in p52 overexpression cells ($n = 4$). **h** cAMP level in HepG2 cells transfected with p52 overexpression plasmid ($n = 6$). Bars represent mean ± SEM values. **i** Intracellular cAMP levels and glucose output in primary hepatocytes transfected with p52 siRNA with or without PDE4B siRNA ($n = 6$). **j** Relative mRNA abundance of *PDE3B* in glucagon stimulated HepG2 cells (100 nM glucagon for 1 h, in vitro) or mice liver tissue (2 mg/kg glucagon for 1 h, in vivo), β-actin levels used as a reference ($n = 6$). PDE phosphodiesterase, HFD high-fat diet, NS normal saline, ns not statistically significant, PBS phosphate buffer solution. Values represent mean ± SEM. Statistical differences were determined by one-way ANOVA. $*p < 0.05$ vs. the control group, $**p < 0.01$ vs. the control group. $^{\#\#}p < 0.01$ vs. AAV8-p52 or p52 plasmid group. Source data are provided as a Source Data file

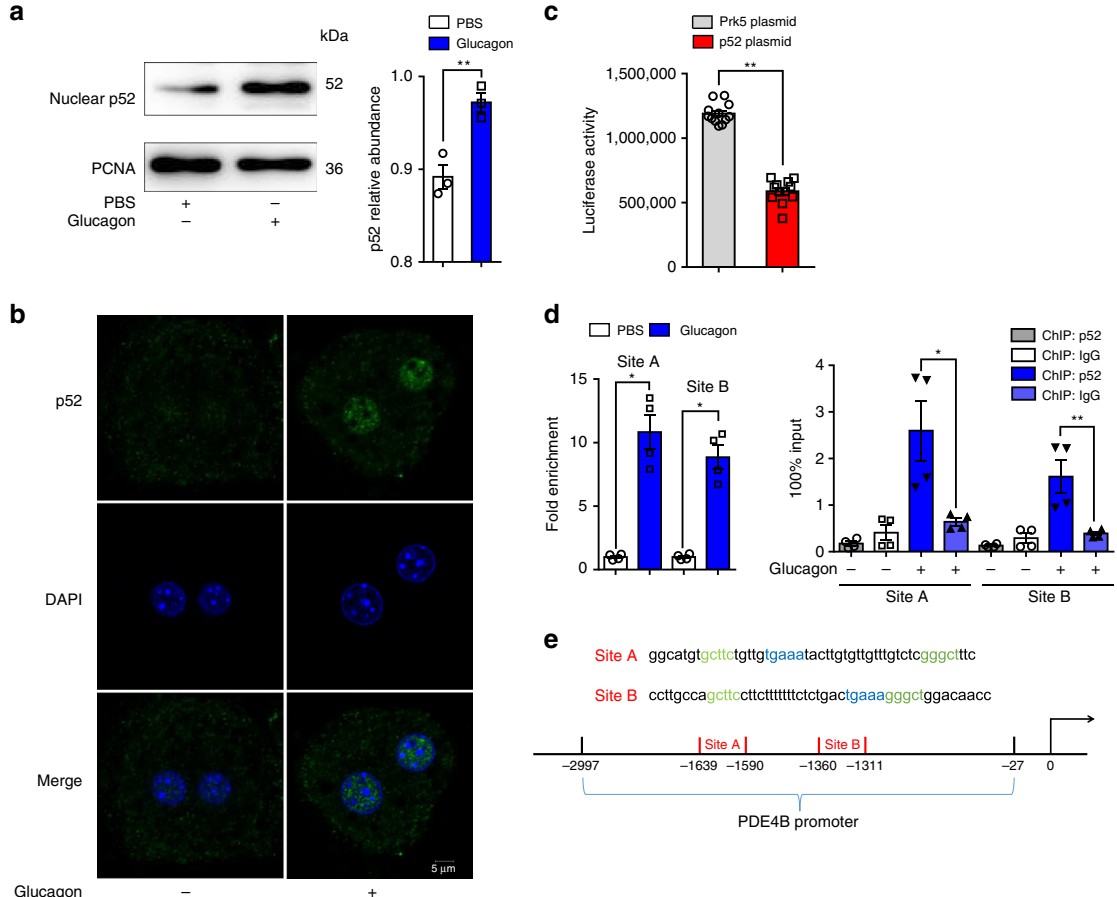

**Fig. 4** p52 binds to the PDE4B promoter and reduces its transcription. **a** The protein level of p52 in cell nuclei when exposed to glucagon (100 nM, 1 h). The PCNA level was used for normalization ($n = 3$). **b** Representative confocal images of primary hepatocytes exposed to glucagon (100 nM, 1 h). Scale bar represents 5 μm. **c** Luciferase reporter assay for inhibition effect of p52 on *PDE4B* gene promoter. The PDE4B luciferase reporter was co-transfected with p52 plasmid (1 μg) in 293 T cells. The luciferase activity was normalized with the internal control (Renilla luciferase, $n = 12$). **d** ChIP analysis to detect p52 binding to the *PDE4B* promoter. HepG2 cells were stimulated by glucagon for 1 h. Equal amounts of chromatin (DNA) were subjected to the ChIP assay with NF-κB2-specific antibody. Mice IgG and protein A/G beads alone were used as negative controls. p52 occupancy of the *PDE4B* promoter is shown relative to background signal with mice IgG control antibody. The ChIP analysis data are shown without normalization as 100% input ($n = 4$). **e** The probable p52-binding sites identified in the PDE4B promoter region. PBS phosphate buffer solution, PDE phosphodiesterase, PCNA proliferating cell nuclear antigen, DAPI 4',6-diamidino-2-phenylindole. The data are presented as the mean ± SEM. Statistical differences between pairs of groups were determined by a two-tailed Student's *t*-test. *$p < 0.05$ vs. control group, **$p < 0.01$ vs. control group. Source data are provided as a Source Data file

Furthermore, under H89-treated conditions, p52 overexpression was not able to restore the gluconeogenesis (Supplementary Fig. 6), indicating that p52 activated gluconeogenesis dependent on the cAMP/PKA pathway. Taken together, glucagon activated p52 through the cAMP/PKA pathway, and the activated p52 in turn augmented cAMP/PKA signaling as a positive feedback loop by inhibiting PDE4B.

Of note, glucagon stimulation had no significant effects on phosphorylated p65 in mice (Supplementary Fig. 7).

**Metformin suppresses hepatic p52 activation in HFD-fed mice**. To determine if pharmacological intervention could inhibit NF-κB2 activation, we examined the effect of metformin on hepatic p52 expression. Metformin attenuated p52 expression in the livers of HFD-fed mice (Fig. 5a), preserved PDE4B induction (Fig. 5b), and prevented cAMP accumulation (Fig. 5c). Metformin had similar effects in glucagon-treated mice (Fig. 5d–f). In primary hepatocytes, metformin inhibited p52 nuclear translocation in response to glucagon (Fig. 5g). These results indicate that metformin suppresses NF-κB2 activation, contributing to reduced cAMP accumulation. Consistent with this, metformin inhibited

PKA activation (Supplementary Fig. 8a), inactivated CREB by dephosphorylation (Supplementary Fig. 8b), and consequently suppressed expression of *G6Pase*, *PEPCK*, and *PGC1-α* (Supplementary Fig. 8c) in the livers of HFD-fed mice. As a result, metformin improved pyruvate tolerance in HFD-fed mice and attenuated the hyperglycemic response in glucagon-treated mice (Supplementary Fig. 8d, e).

To provide evidence that metformin acts through inhibiting p52 to increase PDE4B expression, we overexpressed p52 in mice liver by AAV8-p52, and then detected the hypoglycemic effects of metformin. The results showed that p52 overexpression diminished the inhibitory effects of metformin on glucagon-stimulated gluconeogenesis (Fig. 5h). In vitro, we transfected p52 plasmid in primary hepatocytes and observed that the inhibitory effects of metformin on hepatic glucose production was also diminished (Fig. 5i). Therefore, we can conclude that metformin lowered hyperglycemia at least in part by inhibiting p52 activation.

Ginsenoside is the natural compound most similar to metformin at the signaling pathway level[18]. Interestingly, ginsenoside Rb1, the most abundant active component in ginseng, also inhibited glucagon-induced p52 expression and

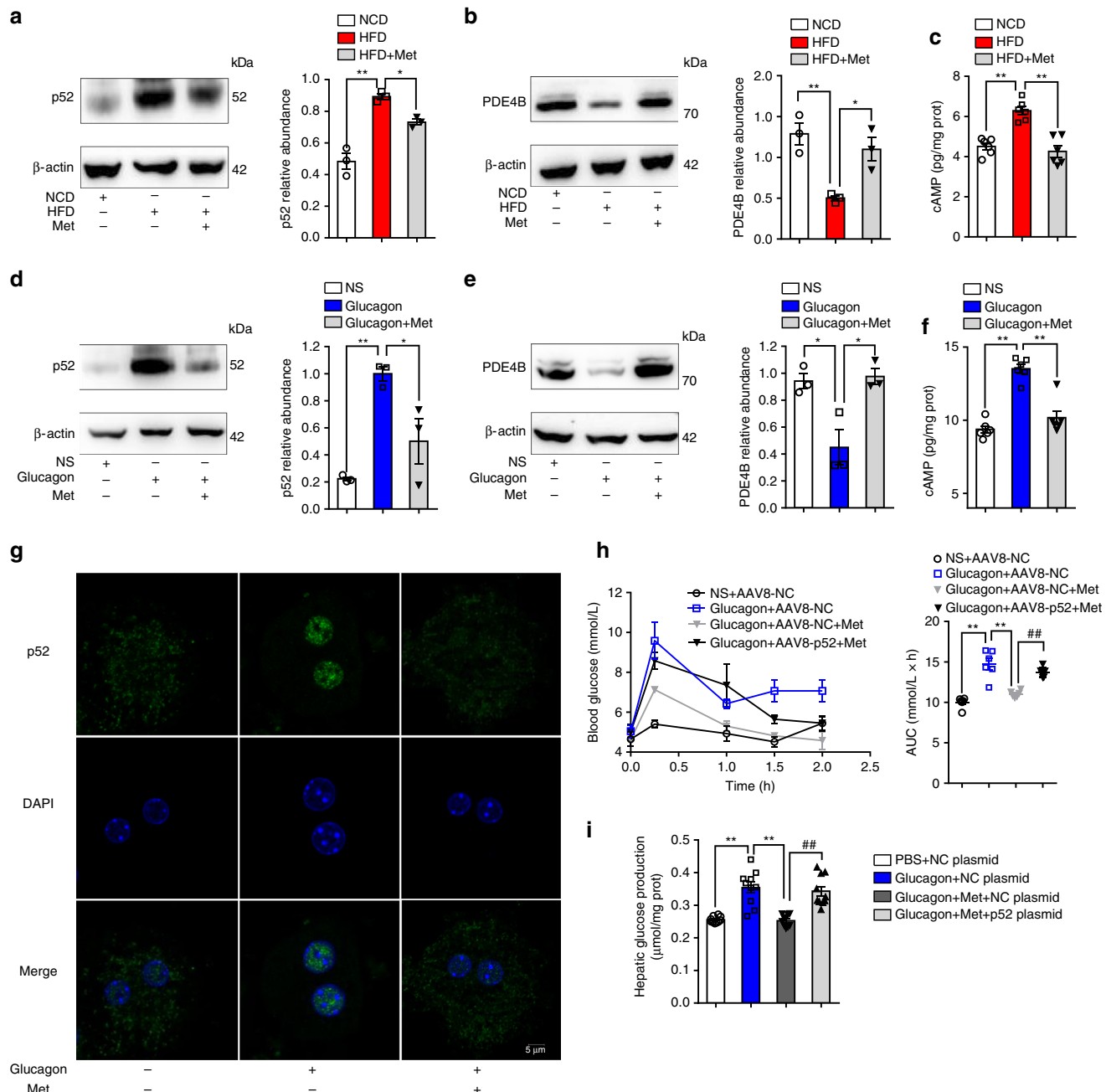

**Fig. 5** Metformin alleviates hyperglycemia by inhibiting p52 activation. **a**, **b** Western blotting analysis of p52 (**a**) and PDE4B (**b**) using liver lysates of mice fed with the indicated diet for 8 weeks. Each lane represents a liver lysate from a different animal. Bar graphs represent the data normalized to β-actin ($n = 3$). Metformin (200 mg/kg/d) was administrated by gavage for 8 weeks. **c** The cAMP levels in the liver of NCD-fed or HFD-fed mice ($n = 6$). **d**, **e** Western blotting analysis of hepatic p52 (**d**) and PDE4B (**e**) in mice injected with 2 mg/kg glucagon ($n = 3$). In total, 200 mg/kg metformin was pre-administrated by gavage 1 h before glucagon injection. **f** Hepatic cAMP levels in glucagon-injected mice ($n = 6$). **g** Representative confocal images of primary hepatocytes exposed to glucagon (100 nM, 1 h) pretreated with metformin (1 mM, for 4 h), or PBS. Scale bar represents 5 μm. **h** Blood glucose curve and AUC for mice that are either treated with AAV8-p52 or AAV8-NC after glucagon injection ($n = 6$). Metformin (200 mg/kg) or normal saline was administrated 1 h before glucagon injection by gavage. **i** Hepatic glucose production in p52 overexpression primary hepatocytes treated with or without 1 mM metformin ($n = 8$). HFD high-fat diet, Met metformin, PDE phosphodiesterase, NCD normal chow diet, NS normal saline, PBS phosphate buffer saline, AUC area under the curve, DAPI 4′,6-diamidino-2-phenylindole, AAV adeno-associated virus. Bars represent mean ± SEM values. Statistical differences were determined by one-way ANOVA. *$p < 0.05$ vs. the control group, **$p < 0.01$ vs. the control group. ##$p < 0.01$ vs. AAV8-p52 or p52 plasmid group. Source data are provided as a Source Data file

nuclear translocation, thereby antagonizing hepatic glucose production (Supplementary Fig. 9a–c). In order to verify that Rb1 exerts its hypoglycemic effects dependent on p52, we overexpressed p52 in mice liver with AAV8-p52 injection. The hypoglycemic effects of Rb1 were diminished in p52

overexpressed mice (Supplementary Fig. 9d). In addition, we transfected p52 plasmid in primary hepatocytes and detected the hepatic glucose production. The inhibitory effects of Rb1 on hepatic glucose production were reversed (Supplementary Fig. 9e).

## Discussion

Dysregulation of hepatic gluconeogenesis is a major contributing factor to the pathogenesis of type 2 diabetes. Until now, the signaling pathways that regulate the key transcription factors of gluconeogenesis remain largely unknown. Better understanding of the precise mechanisms underlying the regulation of gluconeogenesis is crucial for the management of diabetes. Here, we identified a critical role of p52 in glucagon-induced hepatic gluconeogenesis during fasting. The major findings of this work include: (1) p52 was activated by glucagon in livers of HFD-fed mice, and the activated p52 in turn augmented glucagon response as a positive feedback loop; (2) liver-specific p52 knockdown lowered glucagon-stimulated hyperglycemia, while p52 overexpression augmented glucagon response; (3) p52 inhibited *PDE4B* gene promoter activity to promote cAMP accumulation, augmenting glucagon response via cAMP/PKA signaling; (4) metformin and ginsenoside lowered blood glucose at least in part by inhibiting p52 activation. The proposed mechanism is illustrated in Fig. 6. Our data point to a novel therapeutic strategy of targeting p52 activation for the treatment of type 2 diabetes.

Although the role of the canonical NF-κB pathway in type 2 diabetes is well documented, the role of the alternative pathway in type 2 diabetes is just emerging[16,19]. It is generally accepted that p52 is a downstream subunit of NIK kinase in the noncanonical NF-κB signaling. A previous study reported that NIK enhanced hepatic glucose production by increasing CREB stability in obese mice[16]. To test if NIK is required for glucagon-stimulated p100 processing to p52, we silenced NIK in primary hepatocytes using siRNA (Supplementary Fig. 10a). Interestingly, NIK silencing had no significant effect on glucagon-induced p52 activation (Supplementary Fig. 10b), as well as *G6Pase*, *PEPCK*, and *PGC-1α* mRNA expression (Supplementary Fig. 10c) in primary hepatocytes. These results show that glucagon activated p52, at least in part, independent of NIK. Consistent with this, human herpesvirus 8 protein has also been reported to activate p52 independently of NIK[20].

In response to membrane G-protein-coupled receptor, intracellular cAMP generates and serves as a second messenger to mediate cellular responses. However, the generated cAMP can be degraded by PDEs, thereby establishing a negative feedback regulation to prevent excessive cellular responses. Both PDE3B and

PDE4B are expressed in the liver[21,22]. Ke et al. reported that TNF-α initiates canonical NF-κB activation by phosphorylating p65 to inhibit *PDE3B* transcription and impaired insulin sensitivity[21]. Interestingly, we found that glucagon stimulation had no effect on p65 activation and *PDE3B* expression. Activation of p52 transcriptionally suppressed the expression of PDE4B, thereby establishing a positive feedback to amplify the hepatic glucagon response.

It should be noted that p52 knockdown resulted in reduced body weight gain, less fat mass, and decreased lipid deposition in mice. Reduction in adiposity and hepatic steatosis might contribute to the observed glucose phenotypes in the siRNA mice. To exclude the effect of lipid metabolism on glucose phenotypes, we fed mice with HFD for only 5 days instead of 8 weeks. Five day-HFD feeding affected the glucose phenotypes, but had no significant effects on lipid metabolism in terms of body weight, denovo lipogenesis, and lipolysis genes expression (Supplementary Fig. 11). Knockdown of p52 alleviated HFD-induced excessive activation of gluconeogenesis events (Supplementary Fig. 11). In addition, we silenced NF-κB2 using hepatocyte-specific AAV8-shRNA to exclude the off-target effect and further validate the observations. The results showed that liver-specific knockdown had similar effects to siRNA approach (Supplementary Fig. 12). These results excluded the influence of lipid metabolism on glucose phenotypes and proved that p52 knockdown could alleviate glucose disorder independent of the improvement of lipid metabolism. In obesity, the liver may experience integrated stress, such as metabolic stress, oxidative stress, and inflammation. Interestingly, we found the combination simulation of TNF-α, $H_2O_2$, and palmitic acid in primary hepatocytes did not significantly induce increase in gluconeogenic genes expression or hepatic glucose production when compared with glucagon (Supplementary Fig. 13).

Metformin is a first-line therapeutic drug for type 2 diabetes. Ginsenosides, the major bioactive components from ginseng, are reported to be metformin analogues in pharmacological activities[18]. The mechanisms of their actions, however, remains incompletely understood. It has been suggested that metformin suppresses gluconeogenesis by inhibiting adenylyl cyclase enzyme and glucagon-activated gluconeogenic genes transcription[8]. Metformin activates AMPK, contributing to suppression of NF-κB activation[23]. Ginsenoside Rb1 enhanced β-cell insulin secretion via PKA-dependent pathways[24]. We found that metformin and ginsenoside Rb1 suppressed p52 activation in mice, thereby restraining the hepatic glucagon response. In addition, p52 overexpression diminished the inhibitory effects of metformin and ginsenoside Rb1 on glucagon-stimulated gluconeogenesis in mice and in primary hepatocytes. These findings expose new insights into the action of metformin to reduce hepatic glucose output and disclose an unrecognized mechanism for the role of ginsenoside, a metformin analogue, as an antidiabetic agent.

Our work has some limitations: (1) although we established global and liver-specific p52 knockdown mice model using siRNA and AAV-shRNA transfection technology, further validation will be required using liver-specific p52-knockout mice; (2) oral glucose tolerance was improved in p52 knockdown mice, suggesting a potential role for p52 in insulin resistance that will be characterized in future work; (3) whether or not NIK is involved in p100 cleavage to p52 stimulated by glucagon remains to be further characterized.

In summary, we report that glucagon increases cAMP accumulation to induce p52 activation and suppression of PDE4B expression in hepatocytes, thereby directly augmenting hepatic gluconeogenesis in a positive feedback manner. This finding implicates p52 in the dysregulation of hepatic gluconeogenesis, and suggests that pharmacological inhibition of p52 activation

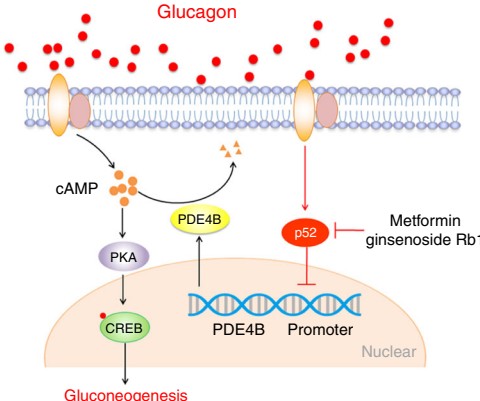

**Fig. 6** NF-κB2 (p52) activation restrains the hepatic glucagon response by preserving PDE4B induction. Glucagon induces p52 activation. Activated p52 binds to *PDE4B* promoter to inhibit its transcription, leading to cAMP accumulation. Accumulated cAMP augmented glucagon response by cAMP/PKA signaling. Metformin and ginsenoside Rb1 inhibited p52 activation to restrain hepatic glucagon response by preserving PDE4B induction

might be a candidate target to reduce excessive hepatic glucose output by restraining the hepatic glucagon response.

## Methods

**Animals and treatments**. Six to eight-week-old male C57BL/6J mice were purchased from Comparative Medical Center of Yangzhou University (Yangzhou, China). After 1 week of acclimation, the mice were fed with a HFD (60% kcal from fat; TP2330; Trophic, China) or a NCD (10% kcal from fat; Xietong Organism, China) for 8 weeks. The mice were raised in a temperature-controlled facility on a 12 h light–dark cycle with free access to food and water. The mice were given metformin (Sigma, America; 200 mg/kg body weight/day) by gavage during the 8 weeks of HFD feeding. Body weights and food intake were recorded every day. Blood was collected after an overnight fast.

The procedures for experiments and animal care were approved by the Institutional Animal Care and Use Committee of China Pharmaceutical University (Nanjing, China). Animal testing and research conforms to all relevant ethical regulations.

**p52 siRNA transfection in vivo**. p52 knockdown mice were created using in vivo transfection technology. In brief, 8-week-old C57BL/6J mice were injected with Entranster-in vivo reagent (Engreen Biosystem, China) carrying p52 small interfering RNA (siRNA) through the caudal vein weekly during HFD feeding. Negative control siRNA oligos were used as controls in chronic and acute experiments. The siRNA oligos are as follows:

Negative control siRNA: UUCUCCGAACGUGUCACGUTT
ACGUGACACGUUCGGAGAATT
p52 siRNA: CCUGAGAGGGAUACCCAAATT
UUUGGGUAUCCCUCUCAGGTT.

**Liver-specific p52 overexpression**. Male C57BL/6J mice at 6–8-week old were purchased from Sino-British SIPPR/BK Lab. Animal Ltd (Shanghai, China). Liver-specific overexpression of p52 in mice was created based on AAV8-p52 plasmid transfection. In brief, after 1 week of acclimation, mice were injected with 7 μL of AAV8-plasmid virus suspension (virus titer > $10^{13}$) blend with 193 μL normal saline. Control mice were injected with AAV8-normal control (AAV8-NC). Three weeks later, mice were randomly divided into different groups for testing fasting blood glucose and glucagon challenge experiments.

**Pyruvate and glucagon tolerance tests**. For pyruvate tolerance tests, fasted mice were injected intraperitoneally with pyruvate (2 g/kg body weight). For glucagon tolerance tests, fasted mice were injected with p52 siRNA oligo via tail vein 48 h before the test. The mice were fasted overnight, and then injected with glucagon (2 mg/kg body weight). During glucagon tolerance test, metformin (200 mg/kg body weight) or ginsenoside Rb1 (50 mg/kg body weight) was administrated by gavage 1 h before glucagon injection. Blood glucose levels were measured using ONE-TOUCH glucose meters (Johnson, America) from tail tips at indicated times.

**Primary hepatocyte isolation and cell culture**. Primary hepatocytes were isolated from 6–8-week-old male C57BL/6J mice using an in situ liver perfusion approach[25]. Isolated primary hepatocytes were cultured in the William E Medium supplemented with 10% (v/v) fetal bovine serum (FBS). HepG2 cell line was obtained from the American Type Culture Collection, and 293T cell line was obtained from Stem Cell Bank, Chinese Academy of Sciences. HepG2 cells and 293T cells were cultured in the Dulbecco's modified eagle medium (DMEM) supplemented with 10% (v/v) FBS.

**In vitro transfection**. For p52 or PDE4B knockdown, HepG2 cells or primary hepatocytes were transfected with p52 or PDE4B siRNA using Lipofectamine® 2000 transfection reagent (ThermoFisher, America) at 50% confluence. Negative control siRNA was used as a control. As for p52 and PDE4B overexpression, cells were transfected with p52 or PDE4B plasmids using the same reagent and methods. Empty vectors (Prk5 or pEX-3) were transfected in control groups.

**Immunofluorescence**. Primary hepatocytes were washed with ice-cold PBS, fixed with 4% paraformaldehyde for 20 min. Then, the cells were incubated with 5% bull serum albumin including 0.2% Triton X-100 to block nonspecific staining, and incubated with anti-NF-κB2 primary antibodies, including 0.2% Triton X-100 staining overnight at 4 °C. The cells were incubated with FITC-labeled goat anti-mouse IgG antibody in the dark at 37 °C for 1 h. They were then incubated with 4',6-diamidino-2-phenylindole in the dark at 37 °C for 10 min. Anti-fade mounting medium was added onto confocal dishes. The cells were mounted on the medium and visualized under a confocal scanning microscope (Carl Zeiss, Germany).

**Quantitative real-time reverse transcription PCR (qRT-PCR)**. The total mRNA was isolated using TRIzol™ Reagent (Invitrogen, America) from the livers of mice, primary hepatocytes or HepG2 cells and cDNAs were synthesized. qRT-PCR was performed on the Roche LightCycler 96 System using the Fast SYBR Green Master Mix (Roche, America). The mRNA expression levels of target genes were normalized to β-actin expression levels. All the primer pairs used are listed in Supplementary Table 1.

**Western blotting**. Liver tissue and cell protein were extracted with a radio-immunoprecipitation assay lysis buffer. Nuclear extracts were prepared using an NE-PER Nuclear Cytoplasmic Extraction Reagent kit (Pierce, America). Protein extracts were separated on 8–10% SDS-PAGE gels and transferred onto nitro-cellulose membranes. Protein expression was visualized by incubating primary antibodies (Supplementary Table 2) overnight at 4 °C followed by the corresponding secondary antibodies. Unprocessed scans of the most important blots in the Supplementary Fig. 14.

**Hepatic glucose production**. HepG2 cells or primary hepatocytes were maintained in the DMEM with 10% FBS. At 50% confluence, the cells were transfected with p52 siRNA or plasmid, PDE4B siRNA or plasmid as mentioned previously. After 24 h, the medium was replaced with Krebs-Ringer HEPES (KRH) buffer to fast the cells for 2 h. After washing with PBS three times, the cells were incubated in KRH buffer supplemented with 10 mM pyruvate, 100 nM glucagon, or with 1 mM metformin and 10 μM ginsenoside Rb1 for 6 h. The cell supernatant was then collected for glucose analysis using the Glucose Assay Kit, and normalized to total cellular protein content.

**Luciferase reporter assay**. Transfection of the pGL3-basic PDE4B promoter and p52 plasmid was conducted in 293T cells using Lipofectamine® 2000. After 48 h, cell lysates were used for luciferase assays using a 96-well luminometer with a dual-luciferase substrate system (Promega, America). A Renilla Luciferase plasmid was used at 0.1 μg/well as an internal control.

**ChIP assays**. Chromatin immunoprecipitation (ChIP) assays were performed using a kit (Magna ChIP HiSens) purchased from Millipore (Temecula, America) according to the manufacturer's instructions. Briefly, chromatin in HepG2 cells pretreated with 100 nM glucagon or vehicle was cross-linked in 1% formaldehyde and subsequently lysed using 1% SDS lysis buffer. Chromatin was fragmented by SFX250 Ultrasonic Cell Disruption System (Branson, America). Soluble chromatin was immunoprecipitated with an NF-κB2 antibody (Santa Cruz Biotechnology, Europe). The de-cross-linked samples were incubated with RNase A and proteinase K. DNA was purified using a Phenol/Chloroform extraction method. The following primers were used during qRT-PCR detection:

Site A: Forward primer GGCATGTGCTTCTGTTGTGA
Reverse primer GAAAGCCCGAGACAAACAA;
Site B: Forward primer CCTTGCCAGCTTCCTTCTT
Reverse primer GGTTGTCCAGCCCTTTCA.

**Statistics**. All data are expressed as mean ± standard error of the mean (SEM). Comparisons between two groups were analyzed by using a two-tailed Student's $t$ test, and those among three or more groups by using one-way analysis of variance (ANOVA). Differences were considered significant at $p < 0.05$. Statistical significance analyses were performed using GraphPad Prism version 7.0.

**Reporting summary**. Further information on research design is available in the Nature Research Reporting Summary linked to this article.

## Data availability

The authors declare that the data supporting the findings of this study are available within the article and its Supplementary Information Files, or are available from the corresponding authors on reasonable request. The RNAseq data of the GTEx were downloaded from the Genotype and Phenotypes database[17], accession number: phs000424.v7.p2. The use of this data set was approved by The National Institutes of Health (NIH). The Project ID was 11502, and the Request ID was 46303–1. The source data underlying Fig. 1a–g, 2a, c–f, h, i, 3a–j, 4a, c, d, and 5a–f, h, i and Supplementary Figs. 1a, b, 2a, b, 3a–d, 4a, b, 5a–e, h, i, 6, 7, 8b–e, 9a, b, d, e, 10a–c, 11a–g, 12a–g, and 13a–d are provided as a Source Data file.

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

## Acknowledgements

We gratefully acknowledge Dr. Brendon Boyce for editing the paper. We thank Prof. Zhengdong Zhang, Prof. Meilin Wang, and Dr. Mulong Du for applying for GTEx data from NIH. This work was supported by the National Natural Science Fund of China for Distinguished Young Scholars (No. 81825023), National Natural Science Foundation of China (Nos 81603353, 91639115, and 81421005), and Natural Science Foundation of Jiangsu Province (BK20160762).

## Author contributions

Q.L., L.-W.Q., and P.L. conceived and designed the experiments. W.-S.Z., Q.L., A.P., X.Z., and A.Y. performed the experiments. G.-X.M. and W.-S.Z. analyzed the data. Q.L., B.-L. L., and W.-S.Z. wrote the paper. L.-W.Q. and P.L. improved the paper. All authors contributed to the discussion of the results and paper corrections.

## Competing interests

The authors declare no competing interests.
