## [Peer Review File · Nature Communications]

Reviewers' comments:

Reviewer #1 (Remarks to the Author):

Zhang et al reported that p52 acts downstream of glucagon receptors to suppress PDE4B expression, thereby enhancing the PKA pathway and the hepatic gluconeogenesis. The authors characterized the p52/PDE4B pathways both in vivo and in hepatocyte cultures, and the findings are interesting and potentially important. However, there are several technical questions complicating data interpretation. Notably, the p65 and NIK pathways have been reported to regulate hepatic PDE activity. This report further confirms and extends the previous reports.

1. It is unclear in the figures and figure legends whether scramble siRNA was used as control in the results and data interpretation.
2. siRNA-treated mice appear to have reduced fat content, which might affect the metabolic phenotypes. This possibility has not been adequately addressed.
3. It is unclear how glucagon and p52 affect PKA levels in the liver (Fig. 3). It is also unclear whether and to what extents the changes in PKA levels contribute to the observed phenotypes.
4. It is also unclear how p52 suppresses PDE4B transcription.

Reviewer #2 (Remarks to the Author):

This study by Zhang et al explores the role of p52 in regulating hepatic gluconeogenesis downstream of glucagon signaling. Increases in p52 in rodent disease models and the role of the non-canonical NFκB pathway has been implicated in previous studies (Nat Med. 2012 Jun; 18(6): 943–949) which affects the novelty of this study.

Major Concerns:

- 1) Use of repeat siRNA injections in vivo often leads to inflammation and may have off target effects. Alternative loss of function techniques would strengthen this manuscript.
- 2) The role of p52 in normal physiological glucagon response (fasting) should be explored and biology of p52 should not be solely inferred from experiments using super physiological glucagon concentrations.
- 3) Does p52 directly interact with the PDE4B promoter? ChIP experiments should be performed to provide further mechanistic insight.
- 4) Hepatic glucose production/gluconeogenesis is not directly measured in this study in vivo and is only correlated with gene targets and a PTT. Tracer studies are should be performed to directly assess the role of p52 and hepatic glucose production
- 5) Does p52 affect glycogen storage/breakdown? This is another important downstream consequence of hepatic glucagon signaling.
- 6) Can glucagon receptor blockade reduce p52 activation?

Minor concerns:

Glucagon potently activates glycogenolysis which and is an Important aspect of hepatic glucose production. This should be mentioned in the main text.

Reviewer #3 (Remarks to the Author):

Comments to the authors and editor:

In the present manuscript, the authors studied the role of the NF- κ B transcription factor p52 in hepatic glucagon response. They observed that p52 protein is induced during HFD in the liver of mice and that in vivo knockdown of p52 lowered fasting blood glucose levels and improved both glucose and pyruvate tolerance in mice treated with this diet. Moreover, it prevented glucagon-mediated increase in blood glucose in normal chow fed mice. In vivo, p52 KD preserved HFD- and glucagon-mediated PDE4B induction and leading to lower of cAMP production. CREB phosphorylation was also decreased in liver of HFD treated p52 KD mice, as well as, the expression of key target genes involved in gluconeogenesis, such as G6pase, PEPCK, and PGC-1 α . The same effect was observed in mice stimulated with glucagon. In vitro, glucagon and activators of cAMP signaling increased p52 levels and nuclear translocation. Additionally, p52 overexpression decreased PDE4B expression and increased cAMP formation. Finally, metformin treatment in mice fed with a HFD or glucagon, inhibited p52 activation, restored PDE4B expression and inhibited cAMP formation.

General comments:

Although the role of the canonical NF- κ B pathway in type 2 diabetes (T2D) is well documented, the role of the alternative pathway in T2D is just emerging. Two previous studies (Shen et al Nat Med 2012 and Liu et al Endocrinology 2017) unveiled a role for NIK, the upstream kinase regulating the alternative NF- κ B pathway, on the modulation of liver gluconeogenesis. In their first study, they demonstrated that NIK is activated in liver of obese mice and it promotes glucagon action and hepatic glucose production. They suggested that these effects are at least partially mediated via a direct effect of NIK on the stabilization of CREB. In their recent study, they showed that deletion of NIK in liver attenuates glucagon and pyruvate-stimulated hepatic glucose production. These effects were correlated with increased PDE3B expression and PDE activity and decreased CREB phosphorylation. These previous studies and the present manuscript confirm a role for the alternative NF- κ B pathway in liver gluconeogenesis, via regulation of similar pathways. However, the two previous studies were focused on NIK and did not analyze its downstream signaling, such as p52 activation. Therefore, the present study complements these previous studies adding another layer of regulation of the alternative pathway on liver gluconeogenesis. Moreover, they suggest a novel mechanism by which metformin regulates hepatic glucose output. However, there are few issues that still need to be addressed in this study. Please see below my detailed comments, questions and suggestions:

1. These studies are based on the knockdown of NF- κ B2, the gene that encodes the p100 protein, the precursor of p52. Is the expression of p100 also induced in liver of mice treated with HFD or glucagon? Decreased p100/p52 ratios indicate activation of the alternative NF- κ B signaling. Is the ratio p100/p52 decreased under HFD or after glucagon stimulation in hepatocytes/liver tissue? p100 can inhibit the canonical NF- κ B pathway. Did the authors check if the canonical NF- κ B pathway is activated under HFD or glucagon stimulation in liver and if KD of NF- κ B2 alters the canonical NF- κ B signaling? It is important to verify if the effects observed are purely due to p52 knockdown, since modifications on the NF- κ B canonical signaling could be also playing a role.
2. Did the authors exclude that adipose and/or muscle tissue were KD for p52 after in vivo siRNA treatment? In Suppl Fig 2B, how many injections with siRNA the mice received? The liver was analyzed how many days after siRNA treatment? Please describe the sequence of the control siRNA used and how it was selected.
3. In the present study the authors observed that in HFD or glucagon stimulated mice, p52 KD decreases CREB phosphorylation, indicating that p52 favors CREB signaling in liver. Decreased CREB phosphorylation was also observed in the study of Shen et al Nat Med 2012, in NIK KO mice. They further demonstrate that NIK phosphorylate CREB stabilizing its expression. The expression of total CREB in Figs 3B and 3E do not seem to be changed. Does it mean that in the present experiments NIK is not activated? What are the possible reasons explaining the different results?
4. The authors demonstrate that generation of cAMP and activation of PKA induces p52 expression and nuclear translocation in primary hepatocytes (Fig. 4). However, they have used H89 as a PKA inhibitor. What is the concentration of H89 used? Taking into account that this inhibitor can also inhibit other kinases (Lochner et al Cardiovasc Drug Rev 2006), these results should be confirmed by using other means to inhibit PKA. Of note, the n for the experiments shown in Fig. 4 are missing in the legend, please complete.
5. The authors conclude, by analyzing the experiments performed in Fig. 5 that p52 decreases PDE4B expression and therefore increases cAMP production. However, to confirm that this is indeed the mechanism, they should KD p52 and PDE4B in parallel and verify if this will prevent the decrease in glucagon-stimulated cAMP production observed in p52 KD cells.

6. It was recently shown that NIK modified the expression of PDE3B in hepatocytes. It would be interesting to check if p52 also modifies the expression of this protein.
7. Fig 6. The effects of PDE4B were analyzed using a chemical inhibitor. Results using a siRNA for PDE4B would produce more relevant results.
8. Fig 6F. Western blots showing p52 levels are missing. Is it induced after TNF or PA treatment? The treatment with TNF and PA (time of exposure, concentrations, PA mix) is also not described.
9. Why some in vitro experiments are performed with primary hepatocytes while others are performed in HepG2 cells or even 293T cells? Are the results on HepG2 cells reproducible in primary hepatocytes?
10. The results correlating the effects of metformin and p52 expression are interesting and suggest, but do not prove, that the beneficial effects of the drug may be mediated by p52. Some in vitro experiments in primary hepatocytes could help to support this hypothesis. For example the authors could test if overexpression of p52 could hamper some of the beneficial effects of metformin on glucose output, PDE4B expression, cAMP production etc.
11. The results of Suppl Fig. 5, ginsenoside Rb1, are very preliminary and do not add much to the study they should be removed.

Minor points:

Page 10, line 193: It is not correct to say that the p52 gene was knocked down, is the NF- κ B2 gene that is KD, please correct.

Legend Fig. 6: The concentration of Ro 20-1724 used is missing. The treatment with this inhibitor is not described in the legend of 3B.

A point-by-point response to the reviewers' concerns

We deeply appreciate the first reviewer's comments on our manuscript (NCOMMS-17-17041).

General Comments: *Zhang et al reported that p52 acts downstream of glucagon receptors to suppress PDE4B expression, thereby enhancing the PKA pathway and the hepatic gluconeogenesis. The authors characterized the p52/PDE4B pathways both in vivo and in hepatocyte cultures, and the findings are interesting and potentially important. However, there are several technical questions complicating data interpretation. Notably, the p65 and NIK pathways have been reported to regulate hepatic PDE activity. This report further confirms and extends the previous reports.*

Response: We thank the Reviewer's valuable comments on our manuscript. Base on the reviewer's suggestion, (1) we tested p-p65 expression *in vitro* and *in vivo* when stimulated by glucagon; (2) To exclude the effect of NIK on the effect of p52, we knockdown NIK in HepG2 cells. The results were detailed in each response to the comments.

Comment 1. It is unclear in the figures and figure legends whether scramble siRNA was used as control in the results and data interpretation.

Response: We used universal negative control siRNA as control, which had no homology with the sequence of the target gene. The sequence of the negative control siRNA: UUCUCCGAACGUGUCACGUTT

ACGUGACACGUUCGGAGAATT

The detailed information has been added in methods and figure legends in the Revised Manuscript.

Comment 2. siRNA-treated mice appear to have reduced fat content, which might affect the metabolic phenotypes. This possibility has not been adequately addressed.

Response: We agree with the Reviewer that p52 knock-down impact the fat content, possibly contributing to regulation of hyperglycemia. To exclude the effect of lipid metabolism on glucose phenotypes, we fed mice with HFD for only 5 days instead of 8 weeks. Results showed that 5 day-HFD feeding affected the glucose phenotypes but

had no significant effects on lipid metabolism in terms of body weight, de novo lipogenesis and lipolysis genes expression. Knockdown of p52 alleviated HFD-induced excessive activation of gluconeogenesis events (Supplementary Fig. 3 in the Revised Manuscript). These results excluded the influence of lipid metabolism on glucose phenotypes and proved that p52 knockdown could alleviate glucose disorder independent of the improvement of lipid metabolism. Related information has been added into the Revised Manuscript (page 19).

Supplementary Fig. 10 Knockdown of p52 suppresses gluconeogenesis in short-period HFD mice. Body weight (a), food intake (b) and fasting blood glucose (c) in p52 knockdown mice fed with HFD for 5 days (n=6). (d) Pyruvate tolerance test (PTT, 2 g/kg body weight) in the mice after overnight fasting. AUC is indicated on the right (n=6). (e, f) Gluconeogenesis (e) and lipid metabolism (f) related genes mRNA abundance in liver tissue of the mice fasted overnight (n=6). (g) p52 protein and mRNA level in liver tissue of mice in (a). Liver tissues were collected from the mice after 5 days feeding. NCD: normal chow diet; HFD: high fat diet; AUC: area under curve; G6pase: glucose-6-phosphatase; PEPCK: phosphoenolpyruvate carboxykinase; PGC-1 α : peroxisome proliferator-activated receptor gamma coactivator-1 alpha; SREBP-1: sterol regulatory element binding proteins-1; HMGCR: 3-hydroxy-3-methylglutaryl-coenzyme A reductase; ABCA5: ATP binding cassette subfamily A member 5; FASN: fatty acid synthase; ACC: acetyl CoA carboxylase; HSL: hormone-sensitive lipase; Acsl1: long-chain-fatty-acid-CoA ligase 1; Lipc: hepatic lipase gene; Acadl: acyl-CoA dehydrogenase; PNPL2: recombinant patatin

like phospholipase domain containing protein 2. Bars represent mean \pm SEM values. Statistical differences between pairs of groups were determined by a two-tailed Student's t-test. *: $p < 0.05$ vs. the control group, **: $p < 0.01$ vs. the control group.

Comment 3. It is unclear how glucagon and p52 affect PKA levels in the liver (Fig. 3). It is also unclear whether and to what extents the changes in PKA levels contribute to the observed phenotypes.

Response: We apologize for our confusing marking. All western blot figures of PKA correspond to phospho-PKA substrates but not PKA protein levels. The levels of phospho-(Ser/Thr) PKA substrates reflect cAMP-stimulated activity. The whole gel images of phospho-PKA substrates were provided (Fig. 2b, 2g and Supplementary Fig. 7a in the Revised Manuscript). In fact, glucagon and p52 could activate PKA, but not affect PKA levels in the liver. Otherwise, in response to glucagon, p52 activation further suppressed PDE4B induction and increased cAMP accumulation, then activated PKA to augment gluconeogenesis.

Fig. 2 (b) phospho-PKA substrates protein levels in liver tissue. Results were repeated for three times. Liver tissues were collected from the mice after 8 weeks HFD or NCD feeding.

Fig. 2 (g) Western blotting images of phospho-PKA substrates in liver tissue. Results were repeated for three times. Liver tissues were collected from the mice stimulated by glucagon (2 mg/kg) for 1 h.

Supplementary Fig. 7. (a) Immunoblotting images of phospho-PKA substrates in liver tissue. Results were repeated for three times. Liver tissues were collected from the mice fed with HFD for 8 weeks. 200 mg/kg metformin was administrated by gavage each day during HFD feeding.

Comment 4. It is also unclear how p52 suppresses PDE4B transcription. .

Response: To further explore the impact of p52 on PDE4B transcription, we performed ChIP assays. We found two potential κ B binding sites in the PDE4B promoter region. The association of p52 at PDE4B promoter site A was 10.8-fold higher and at site B was 8.8-fold higher in glucagon-stimulated cells than in control-treated cells (Fig. 4 d, e). Related information has been added into the Revised Manuscript (page 14).

Fig. 4: (d) ChIP analysis to detect p52 binding to the PDE4B promoter. HepG2 cells were stimulated by glucagon for 1 h. Equal amounts of chromatin (DNA) were subjected to ChIP assay with NF- κ B2-specific antibody. Mice IgG and protein A/G beads alone were used as negative controls. p52 occupancy of the PDE4B promoter is shown relative to background signal with mice IgG control antibody. The ChIP analysis data is shown without normalization as 100% input (n=3). (e) The probable p52 binding sites identified in the PDE4B promoter region. PDE: phosphodiesterase; PCNA: proliferating cell nuclear antigen; DAPI: 4',6-diamidino-2-phenylindole. Data are presented as the mean \pm SEM. Statistical differences between pairs of groups were determined by a two-tailed Student's t-test. *: $p < 0.05$ vs. control group, **: $p < 0.01$ vs. control group.

We deeply appreciate the second reviewer's comments on our manuscript (NCOMMS-17-17041).

General Comments: *This study by Zhang et al explores the role of p52 in regulating hepatic gluconeogenesis downstream of glucagon signaling. Increases in p52 in rodent disease models and the role of the non-canonical NFkB pathway has been implicated in previous studies (Nat Med. 2012 Jun; 18(6): 943–949) which affects the novelty of this study.*

Response: We thank the Reviewer's comments on our manuscript. Regarding the novelty of our work, though it is previously described that the non-canonical NF- κ B pathway modulates liver glucagon signaling (Nat Med. 2012 Jun; 18(6): 943–949), a mechanistic delineation of the p52 mediated signaling has previously not be shown. Thus the manuscript adds novel insight to the interplay of glucagon signaling and cytokine mediated NF- κ B2 activation in obesity related glycemic dysregulation. Additionally, we knockdown NIK in HepG2 cells to exclude the effect of NIK on the effect of p52 when stimulated with glucagon. The results were detailed in each response to the comments.

Comment 1. Use of repeat siRNA injections in vivo often leads to inflammation and may have off target effects. Alternative loss of function techniques would strengthen this manuscript.

Response: According to the Reviewer's suggestion, we have silenced hepatic NF- κ B2 in mice using hepatocyte-specific AAV8-shRNA (Supplementary Fig. 3a in the Revised Manuscript). Results showed that p52 liver-specific knockdown attenuated glucagon-stimulated hyperglycemia in mice (Supplementary Fig. 3b in the Revised Manuscript). Consistently, the transcription levels of G6Pase, PEPCCK, and PGC-1 α were down-regulated (Supplementary Fig. 3c in the Revised Manuscript).

In addition, we have overexpressed hepatic NF- κ B2 in mice using hepatocyte-specific AAV8-p52 and silenced hepatic NF- κ B2 in short-term HFD feeding experiments by AAV8-shRNA (Fig. 1f, g in the Revised Manuscript). These results and related methods have been added into the Revised Manuscript (pages 12).

Supplementary Fig. 3 Liver-specific silencing of p52 suppresses hepatic glucagon response in mice. (a) Protein and mRNA level of p52 in liver tissue, (b) Blood glucose in mice injected with 2 mg/kg glucagon at indicated times. *AUC* is indicated on the right (n=6). (c) Relative mRNA abundance of gluconeogenesis genes in liver tissue of mice 1 h after 2 mg/kg glucagon injection (n=6). *AUC*: area under curve. Bars represent mean \pm SEM values. Statistical differences between pairs of groups were determined by a two-tailed Student's *t*-test. *: $p < 0.05$ vs. the control group, **: $p < 0.01$ vs. the control group.

Fig. 1 (f) Fasting blood glucose of liver-specific p52 overexpression mice. (g) Blood glucose curve and *AUC* for mice that are either treated with AAV8-p52 or AAV8-NC after glucagon injection. *AUC*: area under curve. Bars represent mean \pm SEM values. Statistical differences between pairs of groups were determined by a two-tailed Student's *t*-test. **: $p < 0.01$ vs. the control group.

Comment 2. The role of p52 in normal physiological glucagon response (fasting) should be explored and biology of p52 should not be solely inferred from experiments using super physiological glucagon concentrations.

Response: When diabetes happens, the concentration of glucagon is always higher than normal physiological status, leading to excessive hepatic glucagon response and

hyperglycemia. In this work, we used super physiological glucagon concentrations (*Nature* 2013, 494) to mimic glucagon metabolism disorder in obesity and diabetes. When mice were fasted overnight, the physiological glucagon concentrations did not activate p52 (the first group of Fig. 2e in the Revised Manuscript).

Reference: Miller, R. A. et al. Biguanides suppress hepatic glucagon signalling by decreasing production of cyclic AMP. *Nature*. 494, 256-260 (2013).

Comment 3. Does p52 directly interact with the PDE4B promoter? ChIP experiments should be performed to provide further mechanistic insight.

Response: According to the Reviewer’s suggestion, we have performed ChIP assays. We found two potential κ B binding sites in the PDE4B promoter region. The association of p52 at PDE4B promoter site A was 10.8-fold higher and at site B was 8.8-fold higher in glucagon-stimulated cells than in control-treated cells (Fig. 4 d, e). Related information has been added into the Revised Manuscript (page 14).

Fig. 4: (d) ChIP analysis to detect p52 binding to the PDE4B promoter. HepG2 cells were stimulated by glucagon for 1 h. Equal amounts of chromatin (DNA) were subjected to ChIP assay with NF- κ B2-specific antibody. Mice IgG and protein A/G beads alone were used as negative controls. p52 occupancy of the PDE4B promoter is shown relative to background signal with mice IgG control antibody. The ChIP analysis data is shown without normalization as 100% input (n=3). (e) The probable p52 binding sites identified in the PDE4B promoter region. PDE: phosphodiesterase; PCNA: proliferating cell nuclear antigen; DAPI: 4',6-diamidino-2-phenylindole. Data are presented as the mean \pm SEM. Statistical differences between pairs of groups were determined by a two-tailed Student’s t-test. *: $p < 0.05$ vs. control group, **: $p < 0.01$ vs. control group.

Comment 4. Hepatic glucose production/gluconeogenesis is not directly measured in this study in vivo and is only correlated with gene targets and a PTT. Tracer studies are should be performed to directly assess the role of p52 and hepatic glucose production.

Response: We agree with the Reviewer's comment that tracer studies can directly assess the role of p52 and gluconeogenesis. To increase blood glucose, glucagon promotes hepatic glucose output by increasing glycogenolysis and gluconeogenesis and by decreasing glycogenesis and glycolysis in a concerted fashion via multiple mechanisms. In our study, all the experiment *in vivo* were detected after fasting overnight when gluconeogenesis is the main source of blood glucose at that time, but not glycogenolysis. As we know, glucagon triggered gluconeogenesis through induction of three key targets genes, G6Pase, PEPCK, and PGC-1 α . Pyruvate provides a substrate for hepatic gluconeogenesis and is an indicator of hepatic glucose production. Taken together, we used gene targets along with PTT *in vivo* to assess the role of p52 and gluconeogenesis. Meanwhile, we detected both hepatic glucose production and three target genes *in vitro* to further verify our results.

Comment 5. Does p52 affect glycogen storage/breakdown? This is another important downstream consequence of hepatic glucagon signaling.

Response: As the Reviewer commented, glycogen storage/breakdown is another important downstream consequence of hepatic glucagon signaling. So we tested the relative mRNA abundance of Glycogen synthase gene (Gys1) and Glycogen phosphorylase (Pygl) in mice liver. We observed that, after fasted overnight glucagon had no effect on transcription of glycogen synthase and phosphorylase (data shown in Fig. X1a). We also analyzed the glycogen abundance in mice liver. Results showed that glycogen content did not change when stimulated by glucagon (data shown in Fig. X1b) after overnight fasting. In this work, we focused on glucagon-induced excessive gluconeogenesis and all mice were fasted overnight. Results showed that hepatic glycogen was breakdown completely and glucagon was not able to activate glycogenolysis.

Fig. X1 Relative mRNA abundance (a) and glycogen content (b) in mice liver stimulated by glucagon (2 mg/kg, 1 h). All values are denoted as means \pm SEM. Statistical differences between pairs of groups were determined by a two-tailed Student's t-test. *: $p < 0.05$ vs. the control group, **: $p < 0.01$ vs. the control group.

Comment 6. Can glucagon receptor blockade reduce p52 activation?

Response: According to the Reviewer's suggestion, we stimulated primary hepatocytes with glucagon receptor inhibitor adomeglivant and tested p52 activation. We observed an increase in p52 protein level (Supplementary Fig. 4a), which was prevented by adomeglivant (Supplementary Fig. 4). This result indicated that p52 activation was initiated by glucagon.

Supplementary Fig. 4 (a) Relative protein expression in primary hepatocytes stimulated with the glucagon receptor inhibitor, adomeglivant (10 μ M, pre-treated for 2 h) and glucagon (100 nM, 1 h). β -actin levels served as loading controls. All values are denoted as means \pm SEM. Statistical differences between pairs of groups were determined by a two-tailed Student's t-test. *: $p < 0.05$ vs. the control group, **: $p < 0.01$ vs. the control group.

Comment 7. Glucagon potently activates glycogenolysis which and is an Important aspect of hepatic glucose production. This should be mentioned in the main text.

Response: We agree with Reviewer's comment. Besides affecting gluconeogenesis, glucagon regulates blood glucose by affecting glycogen metabolism. Upon glucagon stimulation, PKA phosphorylates activates glycogen phosphorylase kinase to phosphorylate glycogen, resulting in increased glycogenolysis and the production of

glucose 6-phosphate (G-6-P). G-6-P is then converted into glucose by G6Pase, increasing the glucose pool for hepatic output. Fasting time is the primary factor to determine glycogenolysis or glycogenesis, which is the main contributor to the hepatic glucose production. In our experiment, glucagon did not affect transcription of glycogen synthase and phosphorylase after fasted overnight (data shown in Fig. X1b). Meanwhile, the glycogen content did not alter in different groups (data shown in Fig. X1b). We have discussed this in the Revised Manuscript (page 12).

Fig. X1 Relative mRNA abundance (**a**) and glycogen content (**b**) in mice liver stimulated by glucagon (2 mg/kg, 1 h). All values are denoted as means \pm SEM. Statistical differences between pairs of groups were determined by a two-tailed Student's t-test. *: $p < 0.05$ vs. the control group, **: $p < 0.01$ vs. the control group.

We deeply appreciate the third reviewer's comments on our manuscript (NCOMMS-17-17041).

General Comments: *In the present manuscript, the authors studied the role of the NF- κ B transcription factor p52 in hepatic glucagon response. They observed that p52 protein is induced during HFD in the liver of mice and that in vivo knockdown of p52 lowered fasting blood glucose levels and improved both glucose and pyruvate tolerance in mice treated with this diet. Moreover, it prevented glucagon-mediated increase in blood glucose in normal chow fed mice. In vivo, p52 KD preserved HFD- and glucagon-mediated PDE4B induction and leading to lower of cAMP production. CREB phosphorylation was also decreased in liver of HFD treated p52 KD mice, as well as, the expression of key target genes involved in gluconeogenesis, such as G6pase, PEPCK, and PGC-1 α . The same effect was observed in mice stimulated with glucagon. In vitro, glucagon and activators of cAMP signaling increased p52 levels and nuclear translocation. Additionally, p52 overexpression decreased PDE4B expression and increased cAMP formation.*

Finally, metformin treatment in mice fed with a HFD or glucagon, inhibited p52 activation, restored PDE4B expression and inhibited cAMP formation.

General comments:

Although the role of the canonical NF- κ B pathway in type 2 diabetes (T2D) is well documented, the role of the alternative pathway in T2D is just emerging. Two previous studies (Shen et al Nat Med 2012 and Liu et al Endocrinology 2017) unveiled a role for NIK, the upstream kinase regulating the alternative NF- κ B pathway, on the modulation of liver gluconeogenesis. In their first study, they demonstrated that NIK is activated in liver of obese mice and it promotes glucagon action and hepatic glucose production. They suggested that these effects are at least partially mediated via a direct effect of NIK on the stabilization of CREB. In their recent study, they showed that deletion of NIK in liver attenuates glucagon and pyruvate-stimulated hepatic glucose production. These effects were correlated with increased PDE3B expression and PDE activity and decreased CREB phosphorylation. These previous studies and the present manuscript confirm a role for the alternative NF- κ B pathway in liver gluconeogenesis, via regulation of similar pathways. However, the two previous studies were focused on NIK and did not analyze its downstream signaling, such as p52 activation. Therefore, the present study complements these previous studies adding another layer of regulation of the alternative pathway on liver gluconeogenesis. Moreover, they suggest a novel mechanism by which metformin regulates hepatic glucose output. However, there are few issues that still need to be addressed in this study. Please see below my detailed comments, questions and suggestions:

Response: We thank the Reviewer's positive comments. Based on these suggestions, we have done further experiments to support our conclusions. Details are described in each response to the Reviewer.

Comment 1. These studies are based on the knockdown of NF- κ B2, the gene that encodes the p100 protein, the precursor of p52. Is the expression of p100 also induced in liver of mice treated with HFD or glucagon? Decreased p100/p52 ratios indicate activation of the alternative NF- κ B signaling. Is the ratio p100/p52 decreased under HFD or after glucagon stimulation in hepatocytes/liver tissue? p100 can inhibit the canonical NF- κ B pathway. Did the authors check if the canonical NF- κ B pathway is activated under HFD or glucagon stimulation in liver and if KD of NF- κ B2 alters the canonical NF- κ B signaling? It is important to verify if the effects observed are purely due to p52 knockdown, since modifications on the NF- κ B canonical signaling could be also playing a role.

Response: According to the Reviewer's suggestion, we stimulated primary hepatocytes with glucagon. Results showed that glucagon induced p100 phosphorylation and decreased the expression of p100. Meanwhile, the expression of p52 was increased. The ratio of p100/p52 decreased after glucagon stimulation in hepatocytes, indicating activation of the alternative NF- κ B signaling. These results suggested that glucagon induced p100 phosphorylation and then lead the degradation of p100 to generate p52 (Supplementary Fig. 4d, e in the Revised Manuscript).

We then check if the canonical NF- κ B pathway is activated under glucagon stimulation, glucagon stimulation did not phosphorylated p65 in mice (Supplementary Fig. 6 in the Revised Manuscript). Similarly, knockdown of p52 also had no apparently effect on p65 activation. These results showed that glucagon stimulation and knockdown of p52 in liver had no significant effects on the canonical NF- κ B signaling.

Supplementary Fig. 4 Immunoblots of p52 and p100 levels in primary hepatocytes when exposed to glucagon (100 nM, 1 h) pre-treated with or without H89 (20 μ M, 2 h). All values are denoted as means \pm SEM. Statistical differences between pairs of groups were determined by a two-tailed Student's *t*-test. *: $p < 0.05$ vs. the control group, **: $p < 0.01$ vs. the control group.

Supplementary Fig. 6 Glucagon has no effect on canonical NF- κ B signaling. Phosphorylation levels of p65 in mice liver stimulated by glucagon (2 mg/kg glucagon for 1 h). Bars represent mean \pm SEM values. Statistical differences between pairs of groups were determined by a two-tailed Student's *t*-test.

Comment 2. Did the authors exclude that adipose and/or muscle tissue were KD for p52 after in vivo siRNA treatment? In Suppl Fig 2B, how many injections with siRNA the mice received? The liver was analyzed how many days after siRNA treatment? Please describe the sequence of the control siRNA used and how it was selected

Response: To exclude the role of adipose and/or muscle tissue in p52-augmented glucagon response, we silenced hepatic NF- κ B2 using hepatocyte-specific AAV8-shRNA (Supplementary Fig. 3a in the Revised Manuscript). Results showed that p52 liver-specific knockdown attenuated glucagon-stimulated hyperglycemia in mice (Supplementary Fig. 3b in the Revised Manuscript). Consistently, the transcription levels of G6Pase, PEPCK, and PGC-1 α were down-regulated (Supplementary Fig. 3c in the Revised Manuscript). These results and related methods have been added into the Revised Manuscript (pages 12).

Along the same line, to exclude the effect of lipid metabolism on glucose phenotypes, we fed mice with HFD for only 5 days instead of 8 weeks. Results showed that 5

day-HFD feeding affected the glucose phenotypes but had no significant effects on lipid metabolism in terms of body weight, denovo lipogenesis and lipolysis genes expression (Supplementary Fig. 10 in the Revised Manuscript). Knockdown of p52 alleviated HFD-induced excessive activation of gluconeogenesis events (Supplementary Fig. 10 in the Revised Manuscript). These results excluded the influence of lipid metabolism on glucose phenotypes and proved that p52 knockdown could alleviate glucose disorder independent of the improvement of lipid metabolism. Related information has been added into the Revised Manuscript (page 19).

Additionally, we have detailed the Figure Legends according to the Reviewer's suggestion. In Suppl Fig 2B, the mice received 8 injections with siRNA during the 8 weeks HFD feeding (once a week). The liver was analyzed 2 days after siRNA treatment. We used universal negative control siRNA as control, which had no homology with the sequence of the target gene.

The sequence of the negative control siRNA: UUCUCCGAACGUGUCACGUTT
 ACGUGACACGUUCGGAGAATT

Our negative control siRNA is a general sequence which had been widely cited in literature. It comes from nematodes and has no homology with all mammals. Related information has been added into the Revised Manuscript (page 5).

Supplementary Fig. 3 Liver-specific silencing of p52 suppresses hepatic glucagon

response in mice. (a) Protein and mRNA level of p52 in liver tissue, (b) Blood glucose in mice injected with 2 mg/kg glucagon at indicted times. *AUC* is indicated on the right (n=6). (c) Relative mRNA abundance of gluconeogenesis genes in liver tissue of mice 1 h after 2 mg/kg glucagon injection (n=6). *AUC*: area under curve. Bars represent mean \pm SEM values. Statistical differences between pairs of groups were determined by a two-tailed Student's *t*-test. *: $p < 0.05$ vs. the control group, **: $p < 0.01$ vs. the control group.

Supplementary Fig. 10 Knockdown of p52 suppresses gluconeogenesis in short period HFD mice. Body weight (a), food intake (b) and fasting blood glucose (c) in p52 knockdown mice fed with HFD for 5 days (n=6). (d) Pyruvate tolerance test (PTT, 2 g/kg body weight) in the mice after overnight fasting. *AUC* is indicated on the right (n=6). Gluconeogenesis (e) and lipid metabolism (f) related genes mRNA abundance in liver tissue of the mice fasted overnight (n=6). (g) The p52 protein and mRNA level in liver tissue. Liver tissues were collected from the mice after 5 days feeding. *AUC*: area under curve. Bars represent mean \pm SEM values. Statistical differences between pairs of groups were determined by a two-tailed Student's *t*-test. *: $p < 0.05$ vs. the control group, **: $p < 0.01$ vs. the control group.

Comment 3. In the present study the authors observed that in HFD or glucagon stimulated mice, p52 KD decreases CREB phosphorylation, indicating that p52 favors CREB signaling in liver. Decreased CREB phosphorylation was also observed in the study of Shen et al Nat Med 2012, in NIK KO mice. They further demonstrate that

NIK phosphorylate CREB stabilizing its expression. The expression of total CREB in Figs 3B and 3E do not seem to be changed. Does it mean that in the present experiments NIK is not activated? What are the possible reasons explaining the different results?

Response: In our work, p52 bound to PDE4B promoter to inhibit its transcription and promoted cAMP accumulation, thus augmenting the glucagon response through cAMP/PKA/CREB signaling. It suggested that p52 phosphorylated CREB is due to the loop effect, but not directly binding to CREB and further stabilizing it. In addition, Sheng et al (*Nat Med* 2012) used nuclear extracts when tested the effect of NIK on CREB stability in liver. In our study, we tested the total CREB in cell. Moreover, they fed mice with HFD for 13-18 weeks, but we just fed with HFD for 8 weeks. We deduced that the possible reasons of different results are due to the different mechanism, stimulation and extract methods.

*Comment 4. The authors demonstrate that generation of cAMP and activation of PKA induces p52 expression and nuclear translocation in primary hepatocytes (Fig. 4). However, they have used H89 as a PKA inhibitor. What is the concentration of H89 used? Taking into account that this inhibitor can also inhibit other kinases (Lochner et al *Cardiovasc Drug Rev* 2006), these results should be confirmed by using other means to inhibit PKA. Of note, the n for the experiments shown in Fig. 4 are missing in the legend, please complete.*

Response: The concentration of H89 we used was 20 μ M. “n=3” was added in Fig. 4. We have detailed the Figure Legends according to the Reviewer’s suggestion. To explore the underlying mechanisms by which glucagon induced p52 activation, we focused on cAMP/PKA pathway. Three tool compounds, including forskolin, Bt₂-cAMP and H89 were used to activate or inhibit cAMP/PKA in our experiments. Forskolin activates adenylyl cyclase to generate cAMP from ATP; Bt₂-cAMP mimic cellular cAMP; H89 is a PKA inhibitor. Although H89 may also inhibit other kinases, PKA was indeed blocked in our experiment. Therefore, we can conclude that glucagon activated p52 through cAMP/PKA signaling by both positive and negative validation.

Comment 5. The authors conclude, by analyzing the experiments performed in Fig. 5 that p52 decreases PDE4B expression and therefore increases cAMP production. However, to confirm that this is indeed the mechanism, they should KD p52 and PDE4B in parallel and verify if this will prevent the decrease in glucagon-stimulated cAMP production observed in p52 KD cells.

Response: To verify whether or not p52 silencing antagonizes glucagon signaling was dependent on PDE4B, we knocked down both PDE4B and NF- κ B2 genes in primary hepatocytes. We observed that that the inhibitory effects of p52 silencing on cAMP level and hepatic glucose production were blocked by p52 siRNA and PDE4B siRNA co-transfection (Fig. 3i in the Revised Manuscript). These results demonstrated that p52 silencing inhibited gluconeogenesis in a PDE4B-dependent manner. Related information has been added into the Revised Manuscript (page 13).

Fig. 3 (i) Intracellular cAMP levels and glucose output in primary hepatocytes transfected with p52 siRNA with or without PDE4B siRNA (n=6). Bars represent mean \pm SEM values. Statistical differences between pairs of groups were determined by a two-tailed Student's *t*-test. *: $p < 0.05$ vs. the control group, **: $p < 0.01$ vs. the control group.

Comment 6. It was recently shown that NIK modified the expression of PDE3B in hepatocytes. It would be interesting to check if p52 also modifies the expression of this protein.

Response: According to the Reviewer's suggestion, we firstly checked whether the expression of PDE3B changed under glucagon stimulation. When stimulated by glucagon, PDE3B mRNA expression levels did not change significantly *in vitro* or *in vivo* (Fig. 3j). Furthermore, p52 knockdown had no effect on PDE3B expression (Fig. 3j). Taken together, it suggested that p52 activation had no effect on PDE3B in

HepG2 cells when stimulated with glucagon.

Fig. 3 (j) Relative mRNA abundance of PDE3B in glucagon stimulated primary hepatocytes (100 nM glucagon for 1 h, *in vitro*) or mice liver tissue (2 mg/kg glucagon for 1 h, *in vivo*), β -actin levels used as a reference (n=6). Values represent mean \pm SEM. Statistical differences between pairs of groups were determined by a two-tailed Student's t-test. **: $p < 0.01$ vs. the control group.

Comment 7. Fig 6. The effects of PDE4B were analyzed using a chemical inhibitor. Results using a siRNA for PDE4B would produce more relevant results.

Response: According to the Reviewer's suggestion, we knocked down both PDE4B and NF- κ B2 genes in primary hepatocytes instead of a chemical inhibitor. We observed that that the inhibitory effects of p52 silencing on cAMP level and hepatic glucose production were blocked by p52 siRNA and PDE4B siRNA co-transfection (Fig. 3i in the Revised Manuscript). These results demonstrated that p52 silencing inhibited gluconeogenesis in a PDE4B-dependent manner. Related information has been added into the Revised Manuscript (page 13-14).

Fig. 3 (i) Intracellular cAMP levels and glucose output in primary hepatocytes transfected with p52 siRNA with or without PDE4B siRNA (n=6). Bars represent mean \pm SEM values. Statistical differences between pairs of groups were determined by a two-tailed Student's t-test. *: $p < 0.05$ vs. the control group, **: $p < 0.01$ vs. the control group.

Comment 8. Fig 6F. Western blots showing p52 levels are missing. Is it induced after TNF or PA treatment? The treatment with TNF and PA (time of exposure, concentrations, PA mix) is also not described.

Response: In Fig. 6f, we didn't detect the p52 levels after TNF (20 pg/mL for 8 h) or PA (100 μ M for 8 h) treatment. Sheng (*Nat Med* 2012, 18) reported that NF- κ B non-canonical signaling was activated after TNF treatment. But in our work, we focus the status of hyper-glucagon both *in vivo* and *in vitro*. No matter the normal mice or primary hepatocyte, they were all stimulated with glucagon to imitate the status of glucagon disorder. Therefore, we removed these data in the Revised Manuscript to make our work more confocal.

Comment 9. Why some *in vitro* experiments are performed with primary hepatocytes while others are performed in HepG2 cells or even 293T cells? Are the results on HepG2 cells reproducible in primary hepatocytes?

Response: To some experiments, such as the luciferase reporter gene experiment, the plasmid we used is difficult to transfect into primary hepatocytes because of its length and size. So we used 293T cells as the tool cells.

Based on the Reviewer's suggestion, we used primary hepatocytes to confirm some of our results prior performed on HepG2 cells. We observed that glucagon significantly inhibited PDE4B mRNA and protein expression in primary hepatocytes, leading to cAMP accumulation (data shown in Fig. X2). As expected, knockdown of p52 reversed these changes. These results are in consistent with the data in HepG2 cells.

Fig. X2 (a) The mRNA levels of PDE4B in primary hepatocytes transfected with p52 or scrambled siRNA (n=6). Bar graphs represent the levels of genes normalized to β -actin. (b) The protein expression of PDE4B in p52 knocked down primary hepatocytes, β -actin levels served as loading control. (c) cAMP level in primary

hepatocytes transfected with p52 siRNA. Values represent mean \pm SEM. Statistical differences between pairs of groups were determined by a two-tailed Student's *t*-test. *: $p < 0.05$ vs. the control group, **: $p < 0.01$ vs. the control group.

Comment 10. The results correlating the effects of metformin and p52 expression are interesting and suggest, but do not prove, that the beneficial effects of the drug maybe mediated by p52. Some in vitro experiments in primary hepatocytes could help to support this hypothesis. For example the authors could test if overexpression of p52 could hamper some of the beneficial effects of metformin on glucose output, PDE4B expression, cAMP production etc.

Response: According to the Reviewer's suggestion, we overexpressed p52 in mice liver by AAV8-p52, and then detected the hypoglycemic effects of metformin to provide evidence that metformin acts through inhibiting p52 to increase PDE4B expression. Results showed that p52 overexpression diminished the inhibitory effects of metformin on glucagon-stimulated gluconeogenesis (Fig. 5h in the Revised Manuscript). *In vitro*, we transfected p52 plasmid in primary hepatocytes and observed that the inhibitory effects of metformin on hepatic glucose production was also diminished in p52 overexpression cells (Fig. 5i in the Revised Manuscript). Therefore, we can conclude that metformin lowers hyperglycemia at least in part by inhibiting p52 activation. Related information has been added into the Revised Manuscript (page 16).

Fig. 5 (h) Blood glucose curve and *AUC* for mice that are either treated with AAV8-p52 or AAV8-NC after glucagon injection. 200 mg/kg metformin or normal saline was administrated 1 h before glucagon injection by gavage. **(i)** Hepatic glucose production in p52 overexpression primary hepatocytes treated with or without 1 mM metformin. *AUC*: area under curve. Bars represent mean \pm SEM values. Statistical differences between pairs of groups were determined by a two-tailed Student's *t*-test. **: $p < 0.01$ vs. the control group. ###: $p < 0.01$ vs. the p52 overexpression group.

Comment 11. The results of Suppl Fig. 5, ginsenoside Rb1, are very preliminary and do not add much to the study they should be removed.

Response: Ginsenoside is the natural compound most similar to metformin at the signaling pathway level (*Aging 2017; 9: 2245-2268*). In order to verify Rb1 exerts its hypoglycemic effects dependent on p52, we hepatic-specific overexpressed p52 in mice by injection with AAV8-p52. The hypoglycemic effects of Rb1 were diminished in p52 overexpression mice. In addition, we transfected p52 plasmid in primary hepatocytes and detected the hepatic glucose production. The inhibition effects of Rb1 on hepatic glucose production were reversed (Supplementary Fig. 10d, e in the Revised Manuscript). Taken together, ginsenoside Rb1 lowers hyperglycemia at least in part by inhibiting p52 activation. Related information has been added into the Revised Manuscript (page 16-17).

Supplementary Fig. 10 (d) Blood glucose levels of liver-specific p52 overexpression mice pre-administrated with or without 50 mg/kg Rb1 in glucagon challenge test. AUC is indicated on the right (n=6). **(e)** Hepatic glucose production in p52 overexpression primary hepatocytes treated with or without 10 μM Rb1 (n=6). AUC: area under curve. Bars represent mean ± SEM values. Statistical differences between pairs of groups were determined by a two-tailed Student's *t*-test. **: *p* < 0.01 vs. the control group. #: *p* < 0.01 vs. the p52 overexpression group.

Comment 12. Minor points: Page 10, line 193: It is not correct to say that the p52 gene was knocked down, is the NF-κB2 gene that is KD, please correct. Legend Fig. 6: The concentration of Ro 20-1724 used is missing. The treatment with this inhibitor is not described in the legend of 3B.

Response: It is the NF-κB2 gene that is KD and we have corrected it in the Revised Manuscript. The concentration of Ro 20-1724 was 50 μM. To inhibit PDE4B more specificity, PDE4B siRNA was used instead of the chemical inhibitor in the Revised Manuscript (Fig. 3i in the Revised Manuscript).

Fig. 3 (i) Intracellular cAMP levels and glucose output in primary hepatocytes transfected with p52 siRNA with or without PDE4B siRNA (n=6). Bars represent mean \pm SEM values. Statistical differences between pairs of groups were determined by a two-tailed Student's *t*-test. *: $p < 0.05$ vs. the control group, **: $p < 0.01$ vs. the control group.

Reviewers' comments:

Reviewer #1 (Remarks to the Author):

The authors have partially addressed my concerns.

1. It is still confusing with regard to glucagon relation of p52. Glucagon increases NF- κ B2 mRNA levels, indicating transcriptional regulation. Surprisingly, glucagon stimulation decreased NF- κ B2 precursor p100 levels (Suppl Fig. 4e), raising the possibility that glucagon may also regulate p100 cleavages. These dual mechanisms should be clearly tested and described.
2. The authors attempted to address the role of NIK using shRNA in HepG2 cells. First, the authors only assessed NF- κ B2 mRNA levels but not its activation. NIK is known to activate NF- κ B2 (rather than stimulate NF- κ B2 expression). Second, the authors only examined glucagon response, which is unable to model the liver conditions in obesity. It has been extensively documented that in obesity, the liver experiences integrated stress (e.g. metabolic stress, oxidative stress, inflammation). These stresses are known to activate the NIK/NF- κ B2 pathway, presumably even to a higher level than glucagon. Therefore, it is premature to exclude the role of NIK in hepatic p52 activation and subsequent liver responses in obesity, solely based on the HepG2 model and the glucagon response.
3. The main conclusions about endogenous p52 or NIK were relied entirely on a siRNA approach, which may have off-target effects.

Reviewer #2 (Remarks to the Author):

the authors have adequately addressed my concerns

Reviewer #3 (Remarks to the Author):

The authors addressed most of the points raised by the referees.

The manuscript is very much improved.

Minor comments:

- Figure 1F (please provide Western blot showing overexpression of p52 in mice infected with AAV-8)
- Figures 3g-h (PRK5 is which type of control plasmid, empty vector?, please mention in the material and methods part)
- Figures 3i-j (please provide Western blot or qPCR to prove decreased expression of PDE3B or p52 in siRNA transfected cells)

A point-by-point response to the reviewers' concerns

We deeply appreciate the first reviewer's comments on our manuscript (NCOMMS-17-17041A-Z).

Comment 1. The authors have partially addressed my concerns. It is still confusing with regard to glucagon relation of p52. Glucagon increases NF- κ B2 mRNA levels, indicating transcriptional regulation. Surprisingly, glucagon stimulation decreased NF- κ B2 precursor p100 levels (Suppl Fig. 4e), raising the possibility that glucagon may also regulate p100 cleavages. These dual mechanisms should be clearly tested and described.

Response: We tested the p100 cleavages in primary hepatocytes when stimulated with glucagon for different times. Results showed that p100 decreased in a time-dependent manner while p52 rise accordingly (Fig. X1a). Then we used MG132, a cell-permeable proteasome inhibitor, to inhibit proteasome-mediated cleavage of p100. Results showed that when pretreated with MG132, p100 remained unchanged under glucagon stimulation for different times (Fig. X1b). These results indicated that glucagon induced proteasome-mediated cleavage of p100 to p52.

In addition, we detected the change of NF- κ B2 in mRNA level under glucagon stimulation. Results showed that the increased transcriptional levels of NF- κ B2 by glucagon stimulation was diminished when pretreated with MG132. These results indicated that the transcriptional regulation of NF- κ B2 by glucagon was possibly a compensatory effect due to the p100 cleavage.

Taken together, we reasoned that glucagon regulated NF- κ B2 predominantly by increasing the cleavage of p100 to p52 rather than transcription regulation.

Fig. X1 Glucagon induced proteasome-mediated p100 processing. P100 and p52 protein levels in primary hepatocytes stimulated by glucagon at indicated times without (a) or with MG132 (b). MG132 was added into culture medium 1 h before glucagon stimulation. (c) Relative mRNA abundance of *NF- κ B2* stimulated by glucagon for different duration. (d) Transcriptional levels of *NF- κ B2* stimulated by glucagon for different duration pre-treated with MG132. All values are denoted as means \pm SEM. Statistical differences between pairs of groups were determined by a two-tailed Student's *t*-test. *: $p < 0.05$ vs. the control group, **: $p < 0.01$ vs. the control group.

Comment 2. The authors attempted to address the role of NIK using shRNA in HepG2 cells. First, the authors only assessed NF- κ B2 mRNA levels but not its activation. NIK is known to activate NF- κ B2 (rather than stimulate NF- κ B2 expression). Second, the authors only examined glucagon response, which is unable to model the liver conditions in obesity. It has been extensively documented that in obesity, the liver experiences integrated stress (e.g. metabolic stress, oxidative stress, inflammation). These stresses are known to activate the NIK/NF- κ B2 pathway, presumably even to a higher level than glucagon. Therefore, it is premature to exclude the role of NIK in hepatic p52 activation and subsequent liver responses in obesity, solely based on the HepG2 model and the glucagon response.

Response: According to the Reviewer's suggestion, we used NIK silencing primary hepatocytes to test the role of NIK in glucagon-induced NF- κ B2 activation. Results showed NIK silencing had no significant effect on glucagon-induced p52 expression (Supplementary Fig. 9 in the Revised Manuscript), indicating that glucagon triggered p100 cleavage to p52 at least in part independent of NIK. We agree with the Reviewer

that we cannot exclude the role of NIK in hepatic p52 activation. Since this work aims to investigate the role of p52 in glucagon response, the underlying mechanism of NIK-involved p100 cleavage to p52 is not highlighted in this study. We have added this discussion in limitation part (page 17, 19-20).

To test the role of the liver experiences integrated stress, such as metabolic stress, oxidative stress and inflammation in hepatic glucagon response, we stimulated hepatocytes with a combination of TNF- α (10 ng/mL), hydrogen peroxide (100 nM) and palmitic acid (100 μ M). Results showed that the combination simulation did not significantly induce increase in gluconeogenic genes expression and hepatic glucose production, when compared with glucagon. Although glucagon and other obesity-induced integrated stress may activate the non-canonical NF- κ B pathway, our results indicated that glucagon played a more pivotal role in mediating gluconeogenesis than the three integrated stress (metabolic stress, oxidative stress, inflammation) in obesity.

Supplementary Fig. 9 Glucagon activates p52 and gluconeogenesis partly independent on NIK. Relative mRNA abundance of *NIK* (a) in glucagon-stimulated

primary hepatocytes (100 nM, 1 h) transfected with *NIK* siRNA. Primary hepatocytes transfected with NC siRNA were used as control (n=6). **(b)** p52 protein levels in primary hepatocytes stimulated by glucagon (100 nM, 1 h). **(c)** Gluconeogenic genes relative mRNA abundance. NIK: NF- κ B inducing kinase; G6pase: glucose-6-phosphatase; PEPCK: phosphoenolpyruvate carboxykinase; PGC-1 α : peroxisome proliferator-activated receptor gamma coactivator-1 alpha. All values are denoted as means \pm SEM. Statistical differences between pairs of groups were determined by a two-tailed Student's *t*-test. *: $p < 0.05$ vs. the control group, **: $p < 0.01$ vs. the control group.

Fig. X2 Evaluation of hepatic gluconeogenesis stimulated by glucagon or integrated stress. (a-c) Relative mRNA abundance of gluconeogenesis-related genes. **(d)** Hepatic glucose production stimulated by glucagon or a combination of TNF- α (10 ng/mL), hydrogen peroxide (100 nM) and palmitic acid (100 μ M). G6pase: glucose-6-phosphatase; PEPCK: phosphoenolpyruvate carboxykinase; PGC-1 α : peroxisome proliferator-activated receptor gamma coactivator-1 alpha. All values are denoted as means \pm SEM. Statistical differences between pairs of groups were determined by a two-tailed Student's *t*-test. *: $p < 0.05$ vs. the control group, **: $p < 0.01$ vs. the control group.

Comment 3. The main conclusions about endogenous p52 or NIK were relied entirely on a siRNA approach, which may have off-target effects.

Response: To exclude the off-target effects, we silenced NF- κ B2 using hepatocyte-specific AAV8-shRNA in short-term HFD feeding experiment. Results showed that liver-specific knockdown had similar effects to siRNA approach. Five day-HFD feeding affected the glucose phenotypes but had no significant effects on lipid metabolism in terms of body weight, denovo lipogenesis and lipolysis genes expression. Knockdown of p52 alleviated HFD-induced excessive activation of gluconeogenesis events (Supplementary Fig. 11 in the Revised Manuscript). These results proved that p52 knockdown could alleviate glucose disorder independently of the improvement of lipid metabolism. Related information has been added into the

Besides, we have silenced or overexpressed NF- κ B2 using hepatocyte-specific AAV8 approach in glucagon-challenged mice. Results showed that liver-specific knockdown of NF- κ B2 attenuated glucagon-stimulated hyperglycemia (Supplementary Fig. 3b in the Revised Manuscript). Consistently, the transcription levels of G6Pase, PEPCCK, and PGC-1 α were down-regulated (Supplementary Fig. 3c in the Revised Manuscript). p52 liver-specific overexpression increased fasting blood glucose and augmented glucagon-stimulated hyperglycemia in mice (Fig. 1f, g in the Revised Manuscript).

Supplementary Fig. 11 Liver-specific knockdown of p52 suppressed gluconeogenesis in short-period HFD-fed mice. Body weight (a), food intake (b) and fasting blood glucose (c) in p52 liver-specific knockdown mice after HFD-fed for 5 days. AAV8-p52 shRNA was injected 3 weeks before HFD feeding. (d) Pyruvate tolerance test (PTT, 2 g/kg body weight) of mice after short-period HFD feeding. AUC is indicated on the right (n=6). Gluconeogenic (e) and lipid metabolism (f) genes relative mRNA abundance in liver tissue of short-period HFD-fed mice (n=6). (g) Relative mRNA abundance of NF- κ B2 and p52 protein levels in liver tissue. Liver tissues were collected from the mice after 5 days feeding. NCD: normal chow diet;

HFD: high fat diet; *AUC*: area under curve; G6pase: glucose-6-phosphatase; PEPCK: phosphoenolpyruvate carboxykinase; PGC-1 α : peroxisome proliferator-activated receptor gamma coactivator-1 alpha; SREBP-1: sterol regulatory element binding proteins-1; HMGCR: 3-hydroxy-3-methylglutaryl-coenzyme A reductase; ABCA5: ATP binding cassette subfamily A member 5; FASN: fatty acid synthase; ACC: acetyl CoA carboxylase; HSL: hormone-sensitive lipase; Acs11: long-chain-fatty-acid-CoA ligase 1; Lipc: hepatic lipase gene; Acadl: acyl-CoA dehydrogenase; PNPL2: recombinant patatin like phospholipase domain containing protein 2. Bars represent mean \pm SEM values. Statistical differences between pairs of groups were determined by a two-tailed Student's *t*-test. *: $p < 0.05$ vs. the control group, **: $p < 0.01$ vs. the control group.

Supplementary Fig. 3 Liver-specific silencing of p52 suppresses hepatic glucagon response in mice. (a) Relative mRNA abundance of *NF- κ B2* and p52 protein levels in liver tissue of the mice in liver-specific p52 knockdown mice. (b) Blood glucose in mice injected with 2 mg/kg glucagon at indicated times. *AUC* is indicated on the right (n=6). (c) Relative mRNA abundance of gluconeogenesis genes in liver tissue of mice 1 h after 2 mg/kg glucagon injection (n=6). (d) Hepatic p52 protein levels in liver-specific p52 overexpression mice. Liver tissues were collected from the AAV8-p52 mice stimulated by glucagon (2 mg/kg, 1 h). AAV: adeno-associated virus; NS: Normal Saline; *AUC*: area under curve; G6pase: glucose-6-phosphatase; PEPCK: phosphoenolpyruvate carboxykinase; PGC-1 α : peroxisome proliferator-activated receptor gamma coactivator-1 alpha. Bars represent mean \pm SEM values. Statistical differences between pairs of groups were determined by a two-tailed Student's *t*-test. *: $p < 0.05$ vs. the control group, **: $p < 0.01$ vs. the control group.

Fig. 1 (f) Fasting blood glucose of liver-specific p52-overexpressing mice. (g) Blood glucose curve and *AUC* for mice that are either treated with AAV8-p52 or AAV8-NC after glucagon injection. AAV: adeno-associated virus, *AUC*: area under curve. Bars represent mean \pm SEM values. Statistical differences between pairs of groups were determined by a two-tailed Student's *t*-test. **: $p < 0.01$ vs. the control group.

We deeply appreciate the second reviewer's comments on our manuscript (NCOMMS-17-17041A-Z).

General Comments: *The authors have adequately addressed my concerns.*

Response: We thank the Reviewer's positive comments.

We deeply appreciate the third reviewer's comments on our manuscript (NCOMMS-17-17041A-Z).

General Comments: *The authors addressed most of the points raised by the referees.*

The manuscript is very much improved.

Response: We thank the Reviewer's positive comments. Based on these suggestions, we have done further experiments to support our conclusions. Details are described in each response to the Reviewer.

Comment 1. Figure 1F (please provide Western blot showing overexpression of p52 in mice infected with AAV-8).

Response: According to the Reviewer's suggestion, we tested the protein expression of p52 in AAV8-p52 mice liver. Western blot results showed that AAV8-p52 successfully overexpressed p52 in mice liver (Supplementary Fig. 3d). Related results have been added in the Revised Manuscript (pages 6).

Supplementary Fig. 3 (d) p52 protein levels in AAV8-p52 mice liver tissue. Results were repeated for three times. Liver tissues were collected from the p52 overexpression mice stimulated by glucagon (2 mg/kg for 1 h).

Comment 2. Figures 3g-h (PRK5 is which type of control plasmid, empty vector?, please mention in the material and methods part).

Response: Prk5 we used was one type of empty vector, and we have detailed it in methods part of the Revised Manuscript (page 7).

Comment 3. Figures 3i-j (please provide Western blot or qPCR to prove decreased expression of PDE3B or p52 in siRNA transfected cells).

Response: According to the Reviewer's suggestion, we checked the mRNA levels of NF- κ B2 and PDE4B in siRNA transfected cells in Figures 3i-j. Results showed that NF- κ B2 siRNA successfully inhibited glucagon-stimulated increase of NF- κ B2 mRNA abundance (Fig. X3 a). And the PDE4B siRNA also decreased PDE4B expression in mRNA level (Fig. X3 b). These results indicated that PDE4B and p52 were successfully knocked down in Figures 3i-j.

Fig. X3 siRNA successfully inhibited NF- κ B2 and PDE4B gene transcription. Relative mRNA abundance of NF- κ B2 (a) and PDE4B (b) in siRNA transfection primary hepatocytes. PDE: phosphodiesterase. All values are denoted as means \pm SEM. Statistical differences between pairs of groups were determined by a two-tailed Student's t-test. **: $p < 0.01$ vs. the control group.

REVIEWERS' COMMENTS:

Reviewer #1 (Remarks to the Author):

The authors did additional experiments to address my questions, and the manuscript has been improved. Surprisingly, the authors deleted all data to answer how glucagon stimulation activates p52 (e.g. at the transcriptional level as well as the p100-p52 process level).

REVIEWERS' COMMENTS:

Reviewer #1 (Remarks to the Author):

The authors did additional experiments to address my questions, and the manuscript has been improved. Surprisingly, the authors deleted all data to answer how glucagon stimulation activates p52 (e.g. at the transcriptional level as well as the p100-p52 process level).

Response: Considering that the main line of this manuscript is how p52 mediates hepatic gluconeogenesis, but not the activation of p52, so we did not provide these data in the prior revised manuscript. In this Revised Manuscript, we have added the data on how glucagon stimulation activates p52 in Supplementary Fig. 5f-i.